# Health related quality of life and satisfaction with care of stroke patients in Budapest: A substudy of the EuroHOPE project

Ildikó Szőcs[1,2]*, Balázs Dobi[2,3], Judit Lám[4], Károly Orbán-Kis[5], Unto Häkkinen[6], Éva Belicza[4], Dániel Bereczki[1,2], Ildikó Vastagh[1,7]

1 Department of Neurology, Semmelweis University, Budapest, Hungary, 2 MTA-SE Neuroepidemiological Research Group, Budapest, Hungary, 3 Department of Probability Theory and Statistics, Eötvös Loránd University, Budapest, Hungary, 4 Health Services Management Training Centre, Semmelweis University, Budapest, Hungary, 5 Department of Physiology, University of Medicine and Pharmacy of Tirgu-Mures, Targu-Mures, Romania, 6 Finnish Institute for Health and Welfare, Helsinki, Finland, 7 Department of Neurology, Bajcsy-Zsilinszky Hospital, Budapest, Hungary

* szocsiko@gmail.com

## Abstract

### Background

Disadvantaged socioeconomic status is associated with higher stroke incidence and mortality, and higher readmission rate. We aimed to assess the effect of socioeconomic factors on case fatality, health related quality of life (HRQoL), and satisfaction with care of stroke survivors in the framework of the European Health Care Outcomes, Performance and Efficiency (EuroHOPE) study in Hungary, one of the leading countries regarding stroke mortality.

### Methods

We evaluated 200 consecutive patients admitted for first-ever ischemic stroke in a single center and performed a follow-up at 3 months after stroke. We recorded pre- and post-stroke socioeconomic factors, and assessed case fatality, HRQoL and patient satisfaction with the care received. Stroke severity at onset was scored by the National Institutes of Health Stroke scale (NIHSS), disability at discharge from acute care was evaluated by the modified Rankin Score (mRS). To evaluate HRQoL and patient satisfaction with care we used the EQ-5D-5L, 15D and EORTC IN PATSAT 32 questionnaires.

### Results

At 3 months after stroke the odds of death was significantly increased by stroke severity (NIHSS, OR = 1.209, 95%CI: 1.125–1.299, p<0.001) and age (OR = 1.045, 95%CI: 1.003–1.089, p = 0.038). In a multiple linear regression model, independent predictors of HRQoL were age, disability at discharge, satisfaction with care, type of social dwelling after stroke, length of acute hospital stay and rehospitalization. Satisfaction with care was influenced negatively by stroke severity (Coef. = -1.111, 95%C.I.: -2.159- -0.062, p = 0.040), and positively by having had thrombolysis (Coef. = 25.635, 95%C.I.: 5.212–46.058, p = 0.016) and better HRQoL (Coef. = 22.858, 95%C.I.: 6.007–39.708, p = 0.009).

**Data Availability Statement:** All relevant data are within the manuscript and its Supporting Information files.

**Funding:** The work was supported by the EuroHOPE Research Grant [241721], financed by the European Union, belonging into the seventh Framework Program (FP7) of the European Commission (to IS, DB, JL, EB, UH and IV). It was also supported by grants from the National Brain Research Program (KITA-NAP-13-1-2013-0001, to DB), the Higher Education Institutional Excellence Program and the New National Excellence Program (UNKP-17-3, to DB) of the Ministry of Human Resources of the Government of Hungary. The funders had no role in study design, data collection and analysis, decision to publish, or preparation of the manuscript.

**Competing interests:** The authors have declared that no competing interests exist.

## Conclusion

In addition to age, disability, and satisfaction with care, length of hospital stay and type of social dwelling after stroke also predicted HRQoL. Long-term outcome after stroke could be improved by reducing time spent in hospital, i.e. by developing home care rehabilitation facilities thus reducing the need for readmission to inpatient care.

## Introduction

Stroke has a major impact on the quality of life of patients [1, 2], with considerable differences between populations with various socioeconomic background [3]. People of lower socioeconomic status show higher stroke incidence and mortality, higher severity in the acute phase, higher readmission rate to hospital and lose more disability-adjusted life-years [4–7]. Disparities in clinical or behavioral risk factors or differential access to healthcare facilities do not completely explain regional differences in stroke incidence and outcome [8–12].

The impact of differences in social background on post-stroke health-related quality of life (HRQoL) has not been sufficiently evaluated in Central-Eastern European countries. Although there are some data on the consequences of social inequalities on stroke features among neighborhoods within the capital city of Hungary [13], reports evaluating HRQoL data after stroke are scarce from Central-Eastern European countries [14].

The European Health Care Outcomes, Performance and Efficiency (EuroHOPE) project compared health system performance in 6 European countries (Italy, Finland, Sweden, Scotland, the Netherlands and Hungary) [15], evaluating stroke, acute myocardial infarction, hip fracture, breast cancer, and low-birth-weight. In the EuroHOPE study Hungary had the highest stroke incidence, the largest all-cause case fatality 1 year after stroke and the largest regional differences [16, 17].

To improve long-term outcome after stroke, it would be of major importance to identify social groups and features of post-stroke care that are associated with fatality or reduced HRQoL. We previously found worse stroke outcome in patients residing in the poorest district of Budapest compared to those of the wealthiest neighborhood [18, 19], and also found that the lower the annual taxable income, the lower the age at stroke onset in the 23 districts of Budapest [13].

The aim of the present study was to assess the impact of stroke-related, demographic and socioeconomic factors on acute and 3-months case fatality, on HRQoL and on satisfaction with care of patients after stroke.

## Methods

### Study setting and recruitment of participants

Data presented here were collected within the framework of the EuroHOPE project [15], a retrospective study based on national hospital healthcare administrative records in 6 European countries, with the index ischemic stroke between 2006–2008 [16, 20]. The present study was conducted according to Work Package 2 of the EuroHOPE project ("*HRQol protocol and patient satisfaction questionnaire–Stroke*"). We prospectively included patients with ischemic stroke (International Classification Diseases, ICD-10, I63). According to the study protocol, inclusion of consecutive patients with first-ever ischemic stroke was planned up till including 200 patients. That was intended to be an exploratory study with no formal sample size estimation. Exclusion criteria were history of previous ischemic stroke and refusal by the patient to participate in the survey.

Patients were recruited between September 2012 and May 2013, and evaluated at the Department of Neurology of Semmelweis University, a stroke center with a catchment area covering Districts 8 and 9 of Budapest, and the Ráckeve region. The catchment area of this stroke center was defined by geographic availability, i.e. to be the nearest stroke center for the regions of the catchment area. From the socioeconomic point of view, District 8 ranks the last of the 23 districts of Budapest on the list of income; the other regions of the catchment area are in the medium or high income categories. Income constitutes the major socioeconomic difference between the regions in the catchment area. Due to consecutive patient inclusion, our set of patients may be considered representative of ischemic stroke patients of this stroke center's catchment area.

## Study design

Ethical approval was obtained from the Regional and Institutional Committee of Science and Research Ethics of Semmelweis University (no. 98/2012) and written consent was obtained from each patient or from the carer in case of patients who were unable to consent. Initial evaluation of the patients was performed at admission by the physician on duty. The research investigators re-evaluated each case to ascertain that it is indeed an acute first-ever ischemic stroke. As the next step, clinical features, pre-stroke demographic and socioeconomic factors (Table 1) were recorded by a standard questionnaire. Follow-up was performed three months after the onset of stroke by personal visit or by phone interview to assess post-stroke socioeconomic factors, HRQoL and patient satisfaction with care. Length of post-acute hospital stay (i.e. institutional rehabilitation or rehospitalization for any other reason after discharge from

**Table 1. Parameters assessed in the study.**

| | |
|---|---|
| Sociodemographic factors | • age |
| | • gender |
| | • education |
| | • marital status prior and after stroke |
| Socioeconomic factors | • working status prior and after stroke |
| | • dwelling prior to stroke: |
| | ○ location (resident of poorest District 8 or other neighborhood) |
| | ○ conditions (home or institute) |
| | ○ type of social dwelling (alone, with company, with family help, with professional help) |
| | • dwelling after stroke: |
| | ○ conditions |
| | ○ type of social dwelling |
| Clinical features | • pre-stroke clinical factors: degree of assistance needed, pre-stroke depression or dementia, vascular risk factors, prior drug therapy |
| | • strictly stroke related: severity at onset by the NIHSS, disability at discharge from acute care in mRS, stroke type (TOAST classification) |
| | • management related: thrombolysis status, necessity of intensive care or endarterectomy, motor rehabilitation therapy, post-stroke drug therapy, acute and post-acute length of stay (LOS) |
| Outcome measures | • case fatality: acute (inpatient) and at follow-up |
| | • health-related quality of life of survivors at follow-up |
| | • patient satisfaction with care at follow-up |

NIHSS: National Institutes of Health Stroke Scale; mRS: modified Rankin Scale; TOAST: Trial of Org 10172 in Acute Stroke Treatment; LOS: length of stay.

the acute setting) was recorded. Post-acute hospital stay was defined as the number of all inpatient days after discharge from the index hospital admission within the three months of follow-up.

Outcome measures were case fatality during the initial hospitalization and during follow-up, HRQoL and satisfaction with care of the survivors at follow-up.

## Data collection instruments

During admission and at follow-up we have used detailed, standardized questionnaires to assess sociodemographic and socioeconomic factors listed in Table 1.

Initial stroke severity was assessed by the National Institutes of Health Stroke Scale (NIHSS), [21] on admission, whereas disability at discharge was scored by the modified Rankin Scale (mRS), [22]. The NIHSS is a 15-item neurological examination scale used to evaluate the effect of stroke on the level of consciousness, language, neglect, visual-field loss, extraocular movement, motor strength, coordination, dysarthria, and sensory loss. The NIHSS score ranges from 0 to 42, higher scores representing more severe states. The mRS addresses disability ranging from 0 (asymptomatic) to 6 (death). The classification of ischemic stroke was performed using Trial of Org 10172 in Acute Stroke Treatment (TOAST) criteria [23]. According to TOAST, ischemic stroke can be classified into large-artery atherosclerosis, cardioembolism, small-vessel disease, stroke of other determined etiology, and stroke of undetermined etiology.

Stroke and its consequences have significant effects on health related quality of life (HRQoL). We applied indirect methods to evaluate HRQoL in our study with the use of multi-attribute utility (MAU) instruments: by the EuroQoL-5 Dimension-5 Levels (hereafter EQ-5D), developed by the EuroQoL group [24], and by the HRQoL 15D instrument (hereafter 15D) [25, 26]. Both questionnaires are indirect generic instruments to evaluate HRQoL. EQ-5D addresses 5 dimensions: mobility, self-care, usual activities, pain/discomfort and anxiety/depression, grading these on 5 levels and a visual analog scale. The overall HRQoL described by the EQ-5D can be used to derive health state utility values, and has been found to be a valid descriptive system as a generic health outcome measure in patients with acute stroke [27]. The 15D is a comprehensive, self-administered health-related QoL instrument, which consists of 15 dimensions: breathing, mental function, speech, vision, mobility, usual activity, vitality, hearing, eating, elimination, sleeping, distress, discomfort and symptoms, depression and sexual activity. While the EQ-5D is widely used [28], the results might vary depending on the regression method used [29]. EQ-5D seems somewhat less sensitive when it comes to evaluating several chronic diseases [30–33]. The 15D performs better in terms of reliability, discrimination and responsiveness [34], and performs well after critical illness [28]. The agreement between the two utility measures was only moderate [28], and both the mean utilities and the standard deviation differs between EQ-5D and 15D [35]. For these reasons–as recommended by Feeny et al. [36]–we decided to use multiple questionnaires in this study.

We have also assessed the experience of patients with the care received during the hospital stay by EORTC IN-PATSAT32 (hereafter PATSAT), a tool that was developed by the European Organization for Research and Treatment of Cancer [37]. It evaluates 32 items regarding the satisfaction with technical competence, information provision, interpersonal skills, availability, waiting time, access, comfort and overall care perception. The following terms can be used: global patient satisfaction expressed by the PATSAT score of the questionnaire; three subcategories of patient satisfaction (with doctors, with nurses and with services and care organization) and the item number 32 of the questionnaire which is the general satisfaction with care received during the hospital stay of the patient. A higher scale score represents a higher level of satisfaction with care.

## Statistical modeling

The relationship between variables was first assessed independently of any other effects (i.e. without control variables), followed by multivariate analysis.

We have built two types of statistical models, the first using variables related to pre-stroke features, characteristics of the acute stroke, stroke management and post-stroke variables as well. Including stroke related and post-stroke variables in the analysis was especially important when inspecting patient satisfaction with the care received, as the care received is necessarily applied after the onset of stroke. In a supplementary analysis we excluded variables that describe the patients after the acute stroke.

Logistic regression was used to evaluate acute and follow-up fatality. Linear regression was used during the analysis of the aggregated, continuous scores of HRQoL and patient satisfaction. The ordinal elements and scores of HRQoL and patient satisfaction were analyzed with ordered logistic regression. While HRQoL can be defined for the deceased, we chose to create HRQoL-related models only for the survivors, as the imputation of values for deceased patients could have introduced a bias. We also conducted analysis using post-stroke information, which were only available among the survivors.

Linear and logistic regression tables include the intercept, which is just the mean of the dependent variable or odds of the event when the values of the explanatory variables are zero or at the reference value. The ordered logistic regression models include multiple intercepts, but these are omitted from the tables since their value is not important for model interpretation.

In multivariate model building, first all potential covariates were included in the regression model alone to check their effect on the dependent variable. The multiple regression models then were built in the following way: we defined a set of control variables which were included in all models of a given type of dependent variable. Age, sex and NIHSS were control variables in all models. Discharge mRS was also included as control variable in all models that allowed post-stroke information. Additional control variables in the PATSAT models were education and marital status, in the HRQoL models education and in the death models dwelling district (8[th] or not). We then assessed the effect of all other possible covariates one-by-one, controlled by the previously set variables. In the next step, we placed the control variables and all the other variables which in the previous step had a coefficient significant at the 0.1 level into the model. We then checked which variables were significant at the 0.05 level and eliminated those which were not. In this last step we also considered the viability of the model and checked the effect of pre-stroke and post-stroke version and the original and pooled version of the same variables and also interaction terms. We also aimed to synchronize models with similar dependent variables. The reason behind this algorithm is that it allowed us to check the effect of the variables together with the control variables and then all together. This way we were able to discover relationships which might have remained hidden if the variable selection was based only on univariate analysis. Automatized tools such as stepwise regression were considered but discarded due to low patient number.

## Statistical tests

Distribution of continuous variables was checked by the Shapiro-Wilk test. In case of non-normal distribution, non-parametric tests (Mann-Whitney test and Kruskal-Wallis ANOVA) were used for comparisons. Univariate relationship between two continuous variables were analyzed by correlation analysis (Pearson, Kendall, Spearman), Kruskal-Wallis test, and the Mann-Whitney test. The relationship between two discrete variables was assessed using the chi-squared test of independence. The overall significance of a categorical variable with more than 2 levels in a multiple regression model was checked using likelihood-ratio test.

Goodness of fit of regression models was tested using statistical and visual tools. The Hosmer-Lemeshow test and separation plot [38] were used for logistic regression. Shapiro-Wilk test and quantile-quantile plot were used for linear regression. The ordinal models were also inspected by the Hosmer-Lemeshow test. Multicollinearity was checked using the variable inflation factor. R version 3.5.2 was used for data analysis with packages ggplot2, MASS, separationplot, generalhoslem, car, rms.

## Results

### Participant characteristics

In total 200 patients participated in the study. The general description of the study population can be seen in Table 2. Patients from District 8 had more severe strokes (higher NIHSS on

**Table 2. Patient characteristics.**

| Variable | Variable level | Overall | Not District 8 | District 8 | p-value | Response rate* (% of all 200) |
|---|---|---|---|---|---|---|
| N | | **200** | **159** | **41** | | |
| **Age at stroke (mean (SD))** | - | 68.53 (12.86) | 67.67 (12.53) | 71.88 (13.72) | 0.033 | 200 (100%) |
| **Age at acute death (mean (SD))** | - | 75.00 (14.53) | 78.70 (13.30) | 67.60 (15.44) | 0.198 | 15 (7.50%) |
| **Age at follow-up death (mean (SD))** | - | 75.91 (12.28) | 77.25 (11.66) | 72.33 (13.90) | 0.331 | 33 (16.50%) |
| **Sex (%)** | Female | 88 (44.0) | 72 (45.3) | 16 (39.0) | 0.587 | 200 (100%) |
| | Male | 112 (56.0) | 87 (54.7) | 25 (61.0) | | |
| **Years in education (mean (SD))** | - | 10.81 (2.80) | 10.77 (2.71) | 10.95 (3.14) | 0.983 | 183 (91.50%) |
| **Marital status (%)** | With partner | 112 (56.0) | 88 (55.3) | 24 (58.5) | 0.849 | 200 (100%) |
| | Living alone | 88 (44.0) | 71 (44.7) | 17 (41.5) | | |
| **Pre-stroke employment (%)** | Not in employment | 160 (81.6) | 124 (79.5) | 36 (90.0) | 0.193 | 196 (98%) |
| | In employment | 36 (18.4) | 32 (20.5) | 4 (10.0) | | |
| **Post-stroke employment (%)** | Not in employment | 118 (88.1) | 94 (87.0) | 24 (92.3) | 0.684 | 134 (67%) |
| | In employment | 16 (11.9) | 14 (13.0) | 2 (7.7) | | |
| **Social type of pre-stroke dwelling (%)** | Home alone | 64 (32.0) | 54 (34.0) | 10 (24.4) | 0.189 | 200 (100%) |
| | Home with company | 126 (63.0) | 99 (62.3) | 27 (65.9) | | |
| | Institution | 10 (5.0) | 6 (3.8) | 4 (9.8) | | |
| **Social type of post-stroke dwelling (%)** | Home alone | 22 (16.5) | 19 (17.4) | 3 (12.5) | 0.591 | 133 (66.50%) |
| | Home with company | 96 (72.2) | 79 (72.5) | 17 (70.8) | | |
| | Institution | 15 (11.3) | 11 (10.1) | 4 (16.7) | | |
| **Acute fatality (%)** | No | 185 (92.5) | 149 (93.7) | 36 (87.8) | 0.343 | 200 (100%) |
| | Yes | 15 (7.5) | 10 (6.3) | 5 (12.2) | | |
| **Three-month fatality (including acute death) (%)** | No | 167 (83.5) | 135 (84.9) | 32 (78.0) | 0.413 | 200 (100%) |
| | Yes | 33 (16.5) | 24 (15.1) | 9 (22.0) | | |
| **NIHSS on admission (mean (SD))** | - | 7.88 (6.37) | 7.36 (5.87) | 9.88 (7.79) | 0.141 | 200 (100%) |
| **Discharge mRS (mean (SD)) (median (IQR))** | - | 2.68 (1.80) 2 (3) | 2.57 (1.78) 2 (3) | 3.10 (1.84) 3 (4) | 0.111 | 200 (100%) |
| **PATSAT score (mean (SD))** | - | 73.02 (22.38) | 74.22 (22.35) | 67.28 (22.17) | 0.171 | 122 (61%) |
| **EQ-5D utility index (mean (SD))** | - | 0.73 (0.29) | 0.75 (0.28) | 0.65 (0.34) | 0.16 | 126 (63%) |
| **15D utility index (mean (SD))** | - | 0.77 (0.16) | 0.78 (0.16) | 0.73 (0.17) | 0.156 | 124 (62%) |

Tests used: Mann-Whitney test for age variables, education, PATSAT, EQ-5D utility index, 15D utility index and NIHSS, chi-squared test for patient numbers and mRS.

*Response rate may mean data availably or number of events (i.e. death) depending on the variable. The number/percentage of respondents in complex scores and indices indicates the number of patients where these could be calculated. Individual items may have different response rate. In addition, 15D utility indices were calculated using imputation, thus the response rate of the final score may be higher than the one of the individual items.

NIHSS: National Institutes of Health Stroke Scale; mRS: modified Rankin Scale; PATSAT: EORTC IN-PATSAT32 questionnaire, assessing patient satisfaction; EQ-5D: the EuroQOL-5 Dimensions-5 Levels questionnaire; 15D: generic 15 dimensional measure of HRQoL [26]; IQR: interquartile range.

admission) and more severe disability at discharge (higher mRS scores). Despite an older age at stroke, fatal cases of dwellers of District 8 were younger both at the initial hospitalization and at 3 months after stroke, although none of these differences were statistically significant.

## Case fatality

Overall hospital case fatality was 7.5% (12% among District 8 residents and 6% among those living outside District 8). At 3 months after stroke overall case fatality was 16.5% (22% vs 15% among District 8 residents and those living outside District 8). In a multiple logistic regression model admission NIHSS and living alone prior to stroke (vs. living with partner) increased the odds of acute death. No other variables–including the sociodemographic variables—had a statistically significant effect. At follow-up at 3 months, age at stroke onset, and stroke severity by the NIHSS were significant predictors of case fatality, while living alone vs. living with partner lost its significance. For further information, see Table 3. Our study was underpowered to make reliable comparisons between features of inhabitants of the poorer District 8 and other regions.

## Questionnaire completion

Assessment of HRQoL and patient satisfaction at follow-up could have been possible in 81% of the survivors (136 cases alive, able and willing to express themselves, 7 unable—aphasic, demented or comatose, 24 patients declined to answer). The response rates among the 167 patients surviving 3 months were 122 (73%) for PATSAT score, 126 (75%) for EQ-5D utility index and 124 (74%) for 15D utility index. The difference from 136 is due to incomplete answers at item level. The response rate for HRQoL and patient satisfaction questionnaires were almost the same (73%-75%), while initial stroke severity and disability data (i.e. NIHSS and mRS) could be gathered from all 167 survivors. The 15D missing data were imputed at the item level using an SPSS algorithm for those with less than 4 missing items, based on the age and gender of the patients [25, 26].

**Table 3. Multiple logistic regression models for case fatality.**

| Acute fatality (n = 200, McFadden's $R^2$ = 0.287) | | | | | |
|---|---|---|---|---|---|
| **Variables** | **Reference categories** | **Odds-ratio** | **Lower 95% C.I.** | **Upper 95% C.I.** | **p-value** |
| **Intercept** | - | 0.001 | 0.000 | 0.043 | <0.001 |
| **Sex–Male** | Female | 2.724 | 0.688 | 10.786 | 0.154 |
| **Age at stroke** | - | 1.025 | 0.973 | 1.081 | 0.354 |
| **Dwelling location–District 8** | Not District 8 | 1.144 | 0.288 | 4.539 | 0.848 |
| **NIHSS on admission** | - | 1.167 | 1.077 | 1.264 | <0.001 |
| **Living alone** | Living with partner | 4.448 | 1.043 | 18.973 | 0.044 |
| Three-month fatality (n = 200, McFadden's $R^2$ = 0.315) | | | | | |
| **Variables** | **Reference categories** | **Odds-ratio** | **Lower 95% C.I.** | **Upper 95% C.I.** | **p-value** |
| **Intercept** | - | 0.001 | 0.000 | 0.020 | <0.001 |
| **Sex–Male** | Female | 1.406 | 0.520 | 3.803 | 0.502 |
| **Age at stroke** | - | 1.045 | 1.003 | 1.089 | 0.038 |
| **Dwelling location–District 8** | Not District 8 | 0.788 | 0.257 | 2.417 | 0.677 |
| **NIHSS on admission** | - | 1.209 | 1.125 | 1.299 | <0.001 |
| **Living alone** | Living with partner | 2.460 | 0.925 | 6.540 | 0.071 |

NIHSS: National Institutes of Health Stroke Scale.

## Health-related quality of life

**A. Main analysis.**   During the statistical model building we have initially included at least in the univariate analysis all the factors assessed in the study (see Table 1). The control variables were sex, age, education, NIHSS and discharge mRS. The effect of patient satisfaction was also checked in every HRQoL model. In the multiple linear regression model, we have found that out of all the factors assessed (Table 1) the following had a significant independent impact on the HRQoL measured by EQ-5D index at 90-days after stroke: age, mRS at discharge from the acute hospitalization, stroke type, patient satisfaction, acute and post-acute length of hospital stay, and the type of dwelling after stroke (Table 4). These last two features were related but caused no multicollinearity issues during statistical analysis. The effect of employment status prior to stroke and admission source were only marginally significant predictors of the EQ-5D index, while admission from hospital entailed significantly lower 15D utility scores on average.

The association of age and disability (i.e. mRS at discharge) with HRQoL is already known: the older and the more disabled report poorer quality of life. Some effects are worthy of closer inspection, though. TOAST classification influenced HRQoL in the study population. Cardioembolism was associated with better HRQoL compared to large-artery atherosclerosis

**Table 4.  Predictors of EQ-5D utility index in multiple linear regression model.**

| | EQ-5D utility index (n = 109, adjusted $R^2$ = 0.640) | | | | | | |
|---|---|---|---|---|---|---|---|
| Variables | Reference categories | Coefficient | Lower 95% C.I. | Upper 95% C.I. | p-value | Overall p-value | No. of patients |
| **Intercept** | - | 1.008 | 0.673 | 1.344 | <0.001 | | 109 |
| **Sex–Male** | Female | 0.050 | -0.022 | 0.123 | 0.177 | | 61 |
| **Age at stroke** | - | -0.005 | -0.009 | -0.002 | 0.005 | | 109 |
| **Education** | - | 0.009 | -0.004 | 0.023 | 0.174 | | 109 |
| **NIHSS at admission** | - | -0.006 | -0.016 | 0.004 | 0.230 | | 109 |
| **Discharge mRS = 2–5** | mRS = 0–1 | -0.114 | -0.207 | -0.021 | 0.018 | | 64 |
| **TOAST 2** | TOAST 1 | 0.169 | 0.050 | 0.287 | 0.006 | 0.045 | 21 |
| **TOAST 3** | | 0.099 | -0.015 | 0.212 | 0.094 | | 33 |
| **TOAST 4** | | 0.194 | -0.096 | 0.483 | 0.194 | | 2 |
| **TOAST 5** | | 0.046 | -0.073 | 0.164 | 0.452 | | 29 |
| **Employment prior to stroke: not employed** | Employed | 0.110 | -0.001 | 0.220 | 0.056 | | 88 |
| **Admitted from: other hospital** | Admitted from home | -0.189 | -0.385 | 0.008 | 0.063 | 0.142 | 4 |
| **From other institution** | | -0.067 | -0.225 | 0.092 | 0.414 | | 7 |
| **Acute LOS** | - | -0.011 | -0.017 | -0.005 | <0.001 | | 109 |
| **Dwelling after: at home with company** | Dwelling after: home alone | -0.169 | -0.268 | -0.070 | 0.001 | 0.003 | 78 |
| **In institute or with professionals** | | -0.161 | -0.307 | -0.015 | 0.034 | | 12 |
| **Post-acute LOS: 1–5 days** | Post-acute LOS: 0 days | 0.109 | -0.033 | 0.251 | 0.137 | <0.001 | 8 |
| **6–15 days** | | -0.114 | -0.228 | 0.001 | 0.055 | | 13 |
| **16–30 days** | | -0.103 | -0.262 | 0.055 | 0.204 | | 7 |
| **More than 30 days** | | -0.243 | -0.370 | -0.117 | <0.001 | | 13 |
| **PATSAT score** | - | 0.002 | 0.001 | 0.004 | 0.007 | | 109 |

EQ-5D: the EuroQOL-5 Dimensions-5 Levels questionnaire developed by the EuroQoL group, assessing HRQoL; NIHSS: National Institutes of Health Stroke Scale; mRS: modified Rankin Scale; TOAST 1: large-artery atherosclerosis, TOAST 2: cardioembolism, TOAST 3: small-vessel occlusion, TOAST 4: stroke of other determined etiology, TOAST 5: stroke of undetermined etiology; LOS: length of stay; PATSAT: the questionnaire developed by the European Organization for Research and Treatment of Cancer, named EORTC IN-PATSAT32, assessing patient satisfaction.

expressed in EQ-5D, with marginal effect on 15D (see Tables 4 and 5). Out of the socioeconomic factors assessed in this study, the type of social dwelling after stroke proved to be an independent predictor of EQ-5D index, meaning that the HRQoL at follow-up was highest in patients living alone independently in their home, somewhat lower in those at home with company, even lower in case of patients receiving family help; the poorest results were reported in patients dwelling with a professional carer (Fig 1). The length of post-acute hospital stay also proved to be an independent predictor of both the EQ-5D index and the 15D utility score: the longer the duration of post-acute hospitalization, the lower the HRQoL (Fig 2).

We have also inspected the correlation between the scores of different instruments (EQ-5D utility index, 15D utility index, PATSAT score and its subcategories) and we have found that there is a significant positive correlation between each pair (see S4 Table). When examining the sub-item of general patient satisfaction (item 32 of the PATSAT questionnaire), we have also found a significant relationship with both HRQoL indices (S5 Table, Kruskal-Wallis test p-value = 0.019 for EQ-5D and 0.040 for 15D).

## Patient satisfaction with care

Although 81% of the survivors were able and willing to respond to the questionnaires at follow-up, the response rate of PATSAT was 73%. Even less patients chose the categories of poor,

**Table 5. Predictors of 15D utility index in multiple linear regression model.**

| | | 15D utility index (n = 108, adjusted $R^2$ = 0.522) | | | | | |
|---|---|---|---|---|---|---|---|
| Variables | Reference categories | Coefficient | Lower 95% C.I. | Upper 95% C.I. | p-value | Overall p-value | No. of patients |
| Intercept | - | 0.943 | 0.727 | 1.159 | <0.001 | | 108 |
| Sex–Male | Female | 0.037 | -0.010 | 0.085 | 0.126 | | 61 |
| Age at stroke | - | -0.003 | -0.005 | -0.001 | 0.017 | | 108 |
| Education | - | 0.008 | -0.001 | 0.017 | 0.087 | | 108 |
| NIHSS at admission | - | -0.004 | -0.011 | 0.002 | 0.186 | | 108 |
| Discharge mRS = 2–5 | mRS = 0–1 | -0.112 | -0.172 | -0.051 | <0.001 | | 65 |
| TOAST 2 | TOAST 1 | 0.066 | -0.011 | 0.143 | 0.096 | 0.119 | 21 |
| TOAST 3 | | 0.015 | -0.061 | 0.090 | 0.701 | | 31 |
| TOAST 4 | | 0.095 | -0.094 | 0.284 | 0.326 | | 2 |
| TOAST 5 | | -0.021 | -0.098 | 0.056 | 0.591 | | 30 |
| Employment prior to stroke: not employed | Employed | 0.061 | -0.012 | 0.134 | 0.104 | | 87 |
| Admitted from: other hospital | Admitted from home | -0.209 | -0.337 | -0.081 | 0.002 | 0.005 | 4 |
| From other institution | | -0.048 | -0.152 | 0.055 | 0.362 | | 7 |
| Acute LOS | - | -0.004 | -0.008 | -0.001 | 0.027 | | 108 |
| Dwelling after: at home with company | Dwelling after: home alone | -0.107 | -0.171 | -0.042 | 0.002 | 0.004 | 77 |
| In institute or with professionals | | -0.044 | -0.139 | 0.051 | 0.370 | | 12 |
| Post-acute LOS: 1–5 days | Post-acute LOS: 0 days | 0.109 | 0.011 | 0.208 | 0.032 | 0.106 | 7 |
| 6–15 days | | -0.028 | -0.106 | 0.049 | 0.475 | | 12 |
| 16–30 days | | -0.056 | -0.159 | 0.046 | 0.284 | | 7 |
| More than 30 days | | -0.009 | -0.091 | 0.073 | 0.831 | | 13 |
| PATSAT score | - | 0.001 | 0.000 | 0.002 | 0.031 | | 108 |

15D: the 15-dimension questionnaire assessing HRQoL [26]; NIHSS: National Institutes of Health Stroke Scale; mRS: modified Rankin Scale; TOAST 1: large-artery atherosclerosis, TOAST 2: cardioembolism, TOAST 3: small-vessel occlusion, TOAST 4: stroke of other determined etiology, TOAST 5: stroke of undetermined etiology; LOS: length of stay; PATSAT: the EORTC IN-PATSAT32 questionnaire developed by the European Organization for Research and Treatment of Cancer, assessing patient satisfaction.

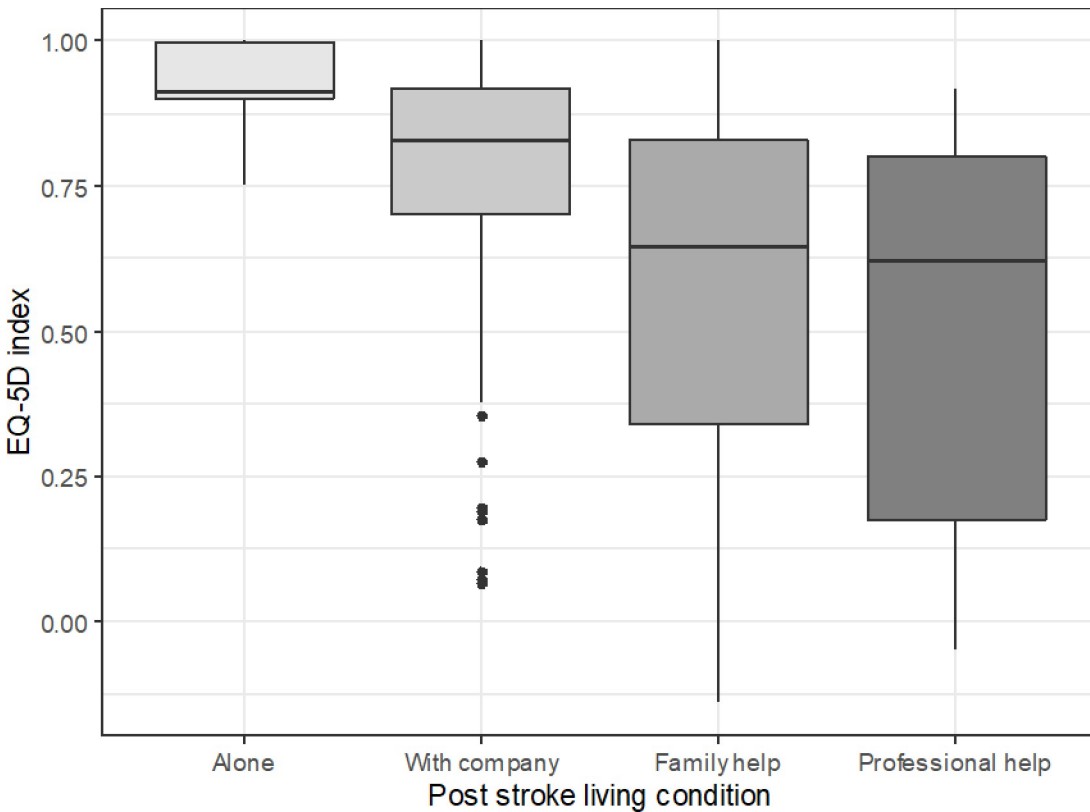

**Fig 1. EQ-5D index by living condition, Kruskal-Wallis chi-squared test p-value < 0.001.** Middle line: median, box borders: IQR, whiskers: range of non-outlier values, points: outliers in Tukey's sense (at least 1.5*IQR distance from the lower or upper quartile), outliers were taken into account when calculating the median and IQR. EQ-5D: the EuroQoL-5 Dimensions-5 Levels questionnaire developed by the EuroQoL group, assessing the health-related quality of life; IQR: interquartile range.

fair or good satisfaction. Thus, when performing ordinal logistic regression analysis, these categories had to be merged due to low patient numbers.

In the *main analyses* of patient satisfaction (see Table 6), the control variables were sex, age, education, NIHSS, discharge mRS and marital status. The effect of HRQoL was also checked in every patient satisfaction model, even if it was not included by the algorithm. Severity of stroke at onset measured by the NIHSS significantly influenced the global patient satisfaction. The EQ-5D index was found in the multiple regression model to be an independent predictor of the global PATSAT score and all its sub-categories at least at a marginally significant level (p<0.1). Thrombolysis also predicted better satisfaction globally, with the care provided by nurses and with organization. Altogether, the multiple linear regression models showed that the patient satisfaction is reduced by stroke severity at admission and is improved by better EQ-5D scores and by having had thrombolysis. The rest of the parameters described in Table 1 showed no effect on global patient satisfaction score or its subcategories. When we performed the multiple ordinal logistic regression model analysis on the question of general patient satisfaction (item 32 of the PATSAT), we have found that both mRS at discharge and EQ-5D index had a marginally significant effect on this item (see Table 6). All patient satisfaction linear regression models showed some deviation from normality (Shapiro-Wilk p<0.05), this somewhat weakens the reliability of the results. There were no significant problems with the fit of the ordinal logistic regression models.

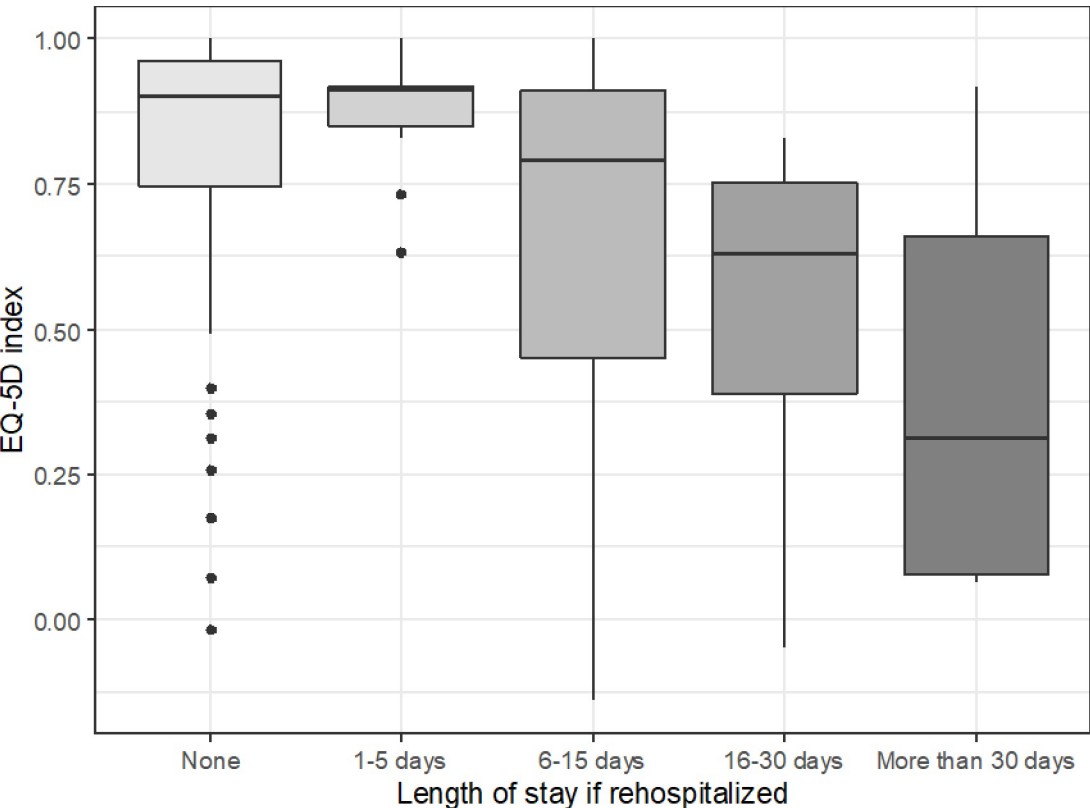

**Fig 2. EQ-5D index by post-acute LOS, Kruskal-Wallis chi-squared test p-value < 0.001.** Middle line: median, box borders: IQR, whiskers: range of non-outlier values, points: outliers in Tukey's sense (at least 1.5*IQR distance from the lower or upper quartile), outliers were taken into account when calculating the median and IQR. EQ-5D: the EuroQOL-5 Dimensions-5 Levels questionnaire developed by the EuroQoL group, assessing the health-related quality of life; LOS: length of stay; IQR: interquartile range.

**B. Analysis excluding post-stroke variables.** We excluded post-stroke variables during this type of model building: we included only age, sex, education, social and anamnestic factors, admission NIHSS and TOAST category. The control variables were sex, age, education and NIHSS. We present the results of these analyses for EQ-5D, 15D and PATSAT in S1–S3 Tables. The multiple linear regression models showed that more advanced age and higher stroke severity were associated with poorer utility indices of both the EQ-5D and 15D (see S1 and S2 Tables). It also showed that ischemic stroke subtype, determined by TOAST criteria might influence 3-month-post-stroke HRQoL. We have found that in this restricted model not only patients who have suffered cardioembolism but also those with small vessel occlusion reported better HRQoL at 3 months after stroke (expressed as higher utility indices of both EQ-5D and 15D) compared to patients who have suffered large vessel atherosclerosis. The stroke subtypes of other determined or undetermined etiology also showed a marginally significant effect on EQ-5D utility index, while this relationship was not detectable in terms of 15D utility index (see S1 and S2 Tables).

We have also found two associations detectable on only one of the two HRQoL instruments used. By the EQ-5D utility index, we have detected that patients who have not been employed prior to their stroke reported higher HRQoL compared to those patients who were employed at time of the stroke. By the 15D utility index, we have seen that patients admitted from another hospital at the onset of the index stroke reported poorer HRQoL compared to their

**Table 6. Multiple linear regression model of the global patient satisfaction (PATSAT score).**

| PATSAT score (n = 111, adjusted $R^2$ = 0.107) | | | | | |
|---|---|---|---|---|---|
| Variables | Reference categories | Coefficient | Lower 95% C.I. | Upper 95% C.I. | p-value |
| Intercept | - | 75.113 | 40.075 | 110.151 | <0.001 |
| Sex: male | Female | 1.530 | -6.679 | 9.738 | 0.716 |
| Age at stroke | - | -0.204 | -0.555 | 0.148 | 0.259 |
| Education | - | -0.508 | -1.968 | 0.953 | 0.497 |
| NIHSS at admission | - | -1.111 | -2.159 | -0.062 | 0.040 |
| mRS on discharge: 2–5 | mRS: 0–1 | 8.410 | -1.714 | 18.535 | 0.107 |
| Marital status: living alone | Living with partner | 2.957 | -5.569 | 11.483 | 0.498 |
| Thrombolysis | No thrombolysis | 25.635 | 5.212 | 46.058 | 0.016 |
| EQ-5D utility index | - | 22.858 | 6.007 | 39.708 | 0.009 |
| PATSAT physicians (n = 115, adjusted $R^2$ = 0.046) | | | | | |
| Variables | Reference categories | Coefficient | Lower 95% C.I. | Upper 95% C.I. | p-value |
| Intercept | - | 78.219 | 39.693 | 116.746 | <0.001 |
| Sex: male | Female | -0.538 | -9.507 | 8.431 | 0.907 |
| Age at stroke | - | -0.237 | -0.625 | 0.152 | 0.235 |
| Education | - | 0.132 | -1.436 | 1.700 | 0.869 |
| NIHSS at admission | - | -0.775 | -1.923 | 0.372 | 0.188 |
| mRS on discharge: 2–5 | mRS: 0–1 | 4.653 | -6.323 | 15.629 | 0.408 |
| Marital status: living alone | Living with partner | 2.894 | -6.375 | 12.163 | 0.542 |
| Thrombolysis | No thrombolysis | 20.890 | -1.843 | 43.624 | 0.075 |
| EQ-5D utility index | - | 18.014 | -0.294 | 36.321 | 0.056 |
| PATSAT nurses (n = 116, adjusted $R^2$ = 0.062) | | | | | |
| Variables | Reference categories | Coefficient | Lower 95% C.I. | Upper 95% C.I. | p-value |
| Intercept | - | 70.984 | 29.478 | 112.491 | 0.001 |
| Sex: male | Female | 5.121 | -4.543 | 14.785 | 0.301 |
| Age at stroke | - | -0.269 | -0.688 | 0.151 | 0.212 |
| Education | - | 0.059 | -1.593 | 1.711 | 0.944 |
| NIHSS at admission | - | -0.696 | -1.933 | 0.541 | 0.272 |
| mRS on discharge: 2–5 | mRS: 0–1 | 3.120 | -8.652 | 14.892 | 0.605 |
| Marital status: living alone | Living with partner | 1.175 | -8.826 | 11.175 | 0.818 |
| Thrombolysis | No thrombolysis | 32.025 | 7.469 | 56.581 | 0.012 |
| EQ-5D utility index | - | 18.512 | -1.166 | 38.189 | 0.068 |
| PATSAT services and care organization (n = 113, adjusted $R^2$ = 0.070) | | | | | |
| Variables | Reference categories | Coefficient | Lower 95% C.I. | Upper 95% C.I. | p-value |
| Intercept | - | 60.287 | 22.805 | 97.769 | 0.002 |
| Sex: male | Female | 4.029 | -4.718 | 12.777 | 0.369 |
| Age at stroke | - | -0.115 | -0.492 | 0.262 | 0.552 |
| Education | - | 0.134 | -1.382 | 1.649 | 0.863 |
| NIHSS at admission | - | -1.035 | -2.155 | 0.086 | 0.073 |
| mRS on discharge: 2–5 | mRS: 0–1 | 6.090 | -4.547 | 16.727 | 0.264 |
| Marital status: living alone | Living with partner | 3.135 | -5.977 | 12.246 | 0.502 |
| Thrombolysis | No thrombolysis | 22.648 | 0.730 | 44.567 | 0.045 |
| EQ-5D utility index | - | 20.266 | 2.469 | 38.064 | 0.028 |
| PATSAT item 32 general satisfaction, ordered logistic regression (n = 112, Nagelkerke $R^2$ = 0.132) | | | | | |
| Variables | Reference categories | Odds-ratio | Lower 95% C.I. | Upper 95% C.I. | p-value |
| Sex: male | Female | 1.404 | 0.685 | 2.878 | 0.354 |
| Age at stroke | - | 0.977 | 0.946 | 1.009 | 0.152 |

*(Continued)*

**Table 6.** (Continued)

| PATSAT score (n = 111, adjusted R² = 0.107) | | | | | |
|---|---|---|---|---|---|
| **Variables** | **Reference categories** | **Coefficient** | **Lower 95% C.I.** | **Upper 95% C.I.** | **p-value** |
| Education | - | 1.025 | 0.901 | 1.165 | 0.710 |
| NIHSS at admission | - | 0.929 | 0.848 | 1.017 | 0.109 |
| mRS on discharge: 2–5 | mRS: 0–1 | 2.177 | 0.924 | 5.132 | 0.075 |
| Marital status: living alone | Living with partner | 1.042 | 0.494 | 2.200 | 0.914 |
| EQ-5D utility index | - | 3.897 | 0.918 | 16.542 | 0.065 |

NIHSS: National Institutes of Health Stroke Scale; mRS: modified Rankin Scale; PATSAT: the EORTC IN-PATSAT32 questionnaire, assessing patient satisfaction; EQ-5D: the EuroQOL-5 Dimensions-5 Levels questionnaire developed by the EuroQoL group, assessing HRQoL.

fellows admitted from their home. The only model that showed significant fitting problems (Shapiro-Wilk p = 0.009) among the HRQoL models was the EQ-5D multiple linear regression model excluding post-stroke variables (S1 Table). The result of the quantile-quantile plot was more dubious. As this was just a sub-analysis and the variable effects are similar in the other models this does not undermine our overall conclusions.

In the supplementary set of analyses (excluding post-stroke variables) we have found that the PATSAT score, satisfaction with services and care organization and the sub-item 32 are significantly influenced by admission NIHSS (see S3 Table). Namely, more severe stroke at admission results in poorer patient satisfaction. We found no significant predictors in these models for satisfaction with physicians and satisfaction with nurses. The control variables were sex, age, education, NIHSS and marital status.

## Discussion

In a consecutive sample of 200 patients with acute ischemic stroke we evaluated predictors of case fatality, health-related quality of life and patient satisfaction. Predictors of acute in-hospital case fatality were stroke severity and living alone prior to stroke. At 3 months after stroke, similarly to previous reports, we also found that case fatality related to initial stroke severity and age. *Health utility index* was diminished after stroke compared to the population norm for EQ-5D for Hungary, and utility indices tended to be lower in the disadvantaged District 8 than in wealthier regions both by EQ-5D and 15D, suggesting that in addition to the consequences of stroke, the general living standard of the patient also affects post-stroke HRQoL. Independent predictors of HRQoL at 3 months after stroke, similarly to other reports, were age, disability at discharge from the acute hospitalization, stroke type, and patient satisfaction. As far as we know, it has not been reported previously that acute and post-acute length of hospital stay after stroke also affect HRQoL. We found evidence that post-acute inpatient management might adversely affect HRQoL: longer than 5 days post-acute LOS was associated with poorer HRQoL. In our study the single independent predictor of all subcategories of *patient satisfaction* was HRQoL. Global patient satisfaction was influenced negatively by initial stroke severity and positively by thrombolysis and HRQoL. According to our knowledge, this positive effect of the provision of thrombolysis on patient satisfaction has not been described before.

### Case fatality

For fatality in the acute phase and at follow-up, the number of patients was low in the different subgroups. This restricted the detailed data analysis and conclusion regarding socioeconomic factors. Nevertheless, we have found that stroke severity and living alone prior to the stroke

increase the odds of dying. These associations are well known. Advanced age and case severity are the most important predictors of early and late case fatality [39, 40]. Living alone was also documented to increase long-term mortality after stroke [41].

## Health-related quality of life

As to the effect of location of residence, we have found that, although dwellers of the socioeconomically disadvantaged District 8 scored less by both health utility measures compared to residents of wealthier regions (Table 1), similarly to case fatality, the number of patients was not sufficient for this difference to reach statistical significance. We have considered the rate of respondents acceptable as it was similar or higher compared to the proportions reported after stroke [42, 43]. Nonetheless we found that the average EQ-5D utility index in both subgroups (0.65±0.34 for District 8 and 0.75±0.28 for other regions) are reduced compared to the population norm of 0.82 [44]. Using EQ-5D the health state utility values after stroke were relatively stable in time: in a systematic review of a total of 199 publications on stroke the median value was 0.63 in studies before 2013 versus 0.65 in studies after 2013 [45]. For 15D, the difference in health utility values between the socioeconomically disadvantaged district and the other areas also did not reach the level of statistical significance in our study (0.73±0.17 for District 8 and 0.78±0.16 for other regions). These values for 15D are within the range reported for stroke patients with or without vision problems (0.73–0.89); [46]. It has been reported that EQ-5D and 15D should not be used interchangeably in economic evaluations after stroke, as the utility scores generated from the two instruments, although correlated well, they differed significantly from each other [47]. Using the same health state utility measures 3 months after intracerebral hemorrhage, predictors for lower HRQoL by both scales were higher NIHSS and older age, with similar ORs for EQ-5D-5L, and 15D [48].

Minimally important differences (MIDs) for health state utilities vary by measure and methods, and are not well established [49]. It has been suggested that for EQ-5D differences among health state utilities of at least 0.036 can be considered clinically important [50]. For EQ-5D, the mean MID among non-stroke patients was 0.074 (range -0.011–0.140) [51]. In a Korean version of the EQ-5D-3L questionnaire, MID in stroke patients ranged from 0.08 to 0.12 [52], and a similar value of 0.10 was reported by Chen et al [53]. Considering the population norms of EQ-5D reported for Hungary (0.82) [44] the mean state utility value for EQ-5D in our patients (0.65 for District 8 and 0.75 for the wealthier regions) is considerably reduced, i.e. HRQoL is obviously adversely affected in stroke survivors 3 months after the acute event with a signal for reduced HRQoL in District 8. As far as 15D is concerned, there is a lack of information on MID values in stroke patients, and also no population norms are known for Hungary. In other patient groups MID for 15D was estimated between 0.01–0.03 [54]. Although by standard statistical methods–similarly to our findings for EQ-5D –the difference between health state utilities by the 15D was not statistically significant, the difference between the mean values of District 8 and other regions (i.e. 0.78–0.73 = 0.05) suggests that HRQoL after stroke is reduced more in socioeconomically deprived regions than in wealthier areas.

It is understandable that graver disability after stroke has a major impact on numerous aspects of the patient's physical, mental, professional and social life which might all be translatable to HRQoL. In the statistical models which excluded post-stroke variables, we have also found that admission stroke severity and advanced age both reduce post-stroke HRQoL [55, 56]. What seems more interesting, though, is the effect of ischemic stroke subtype as to TOAST classification on HRQoL. TOAST classification as a possible predictor of outcome (death or dependency) was extensively studied: only lacunar stroke was found to influence outcome independently from stroke severity measured by the NIHSS [23]. There is scarcity of

data regarding TOAST classification as a predictor of HRQoL. Our finding that cardioembolism was associated with better HRQoL than large-artery atherosclerosis is not in line with previous studies stating that cardioembolism predicts graver physical dependency [57].

Including all variables, we found several independent predictors of HRQoL: age, disability at discharge from acute management, stroke type, patient satisfaction, longer duration of acute and post-acute care and social dwelling type after stroke.

The condition of the patients at discharge from acute management expressed in mRS predicts HRQoL of patients expressed both in EQ-5D and 15D. The more disabled the patients at discharge, the lower the HRQoL of patients at follow-up. This association is quite comprehensible and has been already described [55, 56]. Age is also a well-known predictor of post-stroke HRQoL [58].

The social type of dwelling of patients after stroke also seemed to have an impact on HRQoL at follow-up. Those patients able to live alone independently reported better HRQoL compared to those living with company, and especially compared to those requiring professional help. The finding, that the type of social dwelling itself is associated with a certain HRQoL outcome, independently of the known confounders like stroke severity or disability, is unexpected. The other relationship–onset stroke severity predicting both post-stroke HRQoL and dependence on others—is well-documented [55] and understandable. The social aspects of the dwelling circumstances after stroke might contribute to post-stroke HRQoL above the immediate effects of the stroke related disability on HRQoL.

A strong association between HRQoL and acute length of stay was already described [59, 60]. In our stroke population, above length of acute stay, the duration of post-acute management (i.e. length of stay in rehabilitation or other hospitalization after discharge from the acute care setting) was also associated to poorer post-stroke HRQoL: longer than 5 days (especially more than 30 days) of post-acute hospital-stay had a negative impact on HRQoL. Thus, longer post-acute hospital stay represents not only a costly intervention, but could also contribute to reduced HRQoL of patients. The length of post-acute management can be an expression of graver disability, i.e. definite need for treatment prolongation. On the other hand, it can also be the expression of poorer social background, as patients who are not entirely independent after stroke and are lacking family help will have longer hospital-stays to enable arranging the dwelling conditions necessary after discharge. The association of stroke severity at onset with length of stay and with discharge disposition was already described [61, 62]. Still, length of post-acute hospital-stay was not mentioned yet among the many predictors of post-stroke HRQoL, as far as we know [63, 64]. There is a large body of evidence supporting the benefits of inpatient rehabilitation [65, 66]. What is more important, Stroke Units' Trialists Collaboration [67] states that "people with acute stroke are more likely to survive, return home and regain independence if they receive stroke unit care . . . that can offer a substantial period of rehabilitation if required". In spite of the latter recommendation, a decline in the long-term HRQoL after inpatient rehabilitation of stroke patients was also reported [68]. Early Supported Discharge Trialists bring convincing arguments in favor of shortening the length of inpatient care in selected patients in order to reduce long-term dependency [69, 70]. Further research is needed regarding post-acute management of stroke patients to assess the long-term outcome of inpatient and home-based rehabilitation. Our results suggest that shorter hospital stays followed by well-organized regular outpatient care could contribute to better post-stroke HRQoL. These outpatient interventions could be motor or non-motor rehabilitation sessions, regular home-based check-ups for improving the adherence to secondary prevention, nurses available for administering drugs and assisting the patients in their every-day grooming and medical attendance.

Although non-married status and poorer social support was linked to poorer rehabilitation potential after stroke [71, 72], marital status was not among the predictors of HRQoL in our study.

## Patient satisfaction

In our study, patients reporting better HRQoL also report better satisfaction with the care received and vice versa. There is evidence for each subcategory of patient satisfaction being influenced by the HRQoL, although in certain subcategories the evidence is weak. Our data suggest that patients with better HRQoL are more satisfied with physicians, nurses, organization of care and globally. Patient satisfaction was already known to be one of the most important predictors of HRQoL [73].

In addition to HRQoL, stroke severity also has a major impact on patient satisfaction: more severe strokes are associated with lower global satisfaction. On the other hand, the provision of thrombolysis had a significant positive impact on global patient satisfaction and the subcategory of satisfaction with nurses and care organization. We have not found any description of this association yet.

Patient satisfaction after stroke was not influenced by the rest of the numerous factors assessed. This is consistent with the known difficulty to influence patient satisfaction with direct measures [74, 75] and questions patients' awareness of stroke management possibilities. Supposedly, patients socialized to have an impact on the care received would have valued their own satisfaction in a more differentiated manner. Patient satisfaction improved when they could actively influence the received care [76] and with better patient to nurse ratios [77]. Better patient satisfaction was also linked to outpatient rehabilitation [76–78], but in our study the patients discharged to their home, receiving professional help or living independently, have not valued the stroke care received any more than those discharged to rehabilitation or chronic care facilities.

## Strengths and limitations of the study

The strengths of our study are the consecutive inclusion of patients thus reducing selection bias; the detailed systematic data collection, and the use of dual measures of HRQoL. Our study also has several limitations. First, the sample was relatively small, therefore does not have enough power to evaluate several possible predictors of HRQoL or patient satisfaction. Second, we have taken into account only the presence of pre-stroke depression but did not assess the patients for post-stroke depression, though according to published reports, depression has a significant impact on HRQoL and it can also influence negatively patient satisfaction [79]. Third, we have not assessed HRQoL in the acute phase of stroke, nor in a serial manner, therefore we cannot assess the change in HRQoL. Fourth, among those who did not return for an in-person check-up for the follow-up the rate of less satisfied patients may have been higher, and as the survey was done at 3 months after stroke, recall bias may have had an effect on patient satisfaction. Fifth, although EQ-5D is widely used, this might be a limitation considering its lower sensitivity in evaluating chronic diseases [30, 32, 33, 47]. For this reason, we used 15D as well, adding higher sensitivity and reliability for evaluating HRQoL [30, 32–34, 47].

## Conclusions

Similarly to other studies, we found that initial stroke severity and age are the most important predictors of case fatality at 3 months after stroke. In addition to previously already identified predictors, we found that longer acute and post-acute hospital stay and the type of social dwelling were also associated with HRQoL of stroke survivors. These associations raise the issue

whether long-term outcome could be improved by more intensive inpatient rehabilitation or rather by developing home care facilities. More research is needed regarding the effect of early discharge to patients' home supported by outpatient rehabilitation on HRQoL of the patients.

## Supporting information

**S1 Table. Predictors of EQ-5D utility index in multiple linear regression model excluding post-stroke variables.**
(DOCX)

**S2 Table. Predictors of 15D utility index in multiple linear regression model excluding post-stroke variables.**
(DOCX)

**S3 Table. Predictors of PATSAT score and its subcategories in multiple linear regression and ordered logistic regression models excluding post-stroke variables.**
(DOCX)

**S4 Table. Correlation between EQ-5D utility index, 15D utility index, PATSAT score and its subcategories.**
(DOCX)

**S5 Table. Association between the sub-item of general patient satisfaction (item 32 of the PATSAT questionnaire) and quality of life indices.**
(DOCX)

**S1 Data.**
(XLSX)

## Acknowledgments

The authors are grateful to the stroke patients of the Semmelweis University, Department of Neurology for their active involvement in the study. They are also grateful to the staff of the Department for their helpfulness.

The authors are deeply grateful to Harri Sintonen for providing the 15D questionnaire with its valuation algorithm and above all, for his professional guidance using the tool.

## Author Contributions

**Conceptualization:** Judit Lám, Unto Häkkinen, Éva Belicza, Dániel Bereczki.

**Data curation:** Ildikó Szőcs, Balázs Dobi, Károly Orbán-Kis, Dániel Bereczki.

**Formal analysis:** Ildikó Szőcs, Balázs Dobi, Károly Orbán-Kis, Dániel Bereczki.

**Funding acquisition:** Unto Häkkinen, Éva Belicza, Dániel Bereczki.

**Investigation:** Ildikó Szőcs, Ildikó Vastagh.

**Methodology:** Ildikó Szőcs, Judit Lám, Éva Belicza, Dániel Bereczki, Ildikó Vastagh.

**Project administration:** Ildikó Szőcs, Éva Belicza, Dániel Bereczki.

**Resources:** Éva Belicza, Dániel Bereczki.

**Software:** Balázs Dobi, Éva Belicza.

**Supervision:** Unto Häkkinen, Éva Belicza, Dániel Bereczki.

**Writing – original draft:** Ildikó Szőcs, Balázs Dobi, Károly Orbán-Kis.

**Writing – review & editing:** Ildikó Szőcs, Judit Lám, Unto Häkkinen, Éva Belicza, Dániel Bereczki, Ildikó Vastagh.

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
