## [Decision Letter · Decision Letter 0]

12 May 2020

PONE-D-20-05987

Quality of life and satisfaction with care of stroke patients in Budapest: a substudy of the EuroHOPE project

PLOS ONE

Dear MD Szőcs,

Thank you for submitting your manuscript to PLOS ONE. After careful consideration, we feel that it has merit but does not fully meet PLOS ONE’s publication criteria as it currently stands. Therefore, we invite you to submit a revised version of the manuscript that addresses the points raised during the review process.

Please revise your manuscript according to the reviewers comments.

We would appreciate receiving your revised manuscript by Jun 25 2020 11:59PM. To enhance the reproducibility of your results, we recommend that if applicable you deposit your laboratory protocols in protocols.io, where a protocol can be assigned its own identifier (DOI) such that it can be cited independently in the future. For instructions see: http://journals.plos.org/plosone/s/submission-guidelines#loc-laboratory-protocols

We look forward to receiving your revised manuscript.

Kind regards,

Seana Gall

Academic Editor

PLOS ONE

Journal Requirements:

2. In your Methods section, please provide additional information about the participant recruitment method of the EuroHOPE project and the demographic details of your participants. Please ensure you have provided sufficient details to replicate the analyses such as: a) the recruitment date range (month and year), b) a description of any inclusion/exclusion criteria that were applied to participant recruitment, c) a table of relevant demographic details, d) a statement as to whether your sample can be considered representative of a larger population, e) a description of how participants were recruited, and f) descriptions of where participants were recruited and where the research took place. Moreover, please provide more information on the catchment area and on the characteristics of the District 8.

Reviewers' comments:

Reviewer's Responses to Questions

**Comments to the Author**

1. Is the manuscript technically sound, and do the data support the conclusions?

Reviewer #1: Partly

Reviewer #2: Partly

2. Has the statistical analysis been performed appropriately and rigorously? 

Reviewer #1: No

Reviewer #2: Yes

3. Have the authors made all data underlying the findings in their manuscript fully available?

Reviewer #1: No

Reviewer #2: No

4. Is the manuscript presented in an intelligible fashion and written in standard English?

Reviewer #1: Yes

Reviewer #2: Yes

5. Review Comments to the Author

Reviewer #1: This is an interesting study investigating the impact of social background on health-related quality of life (HRQoL) and satisfaction with care among those with stroke. However, some concerns or comments are provided below for authors to consider.

Abstract

1) ‘Higher severity in the acute phase’: do you mean more severe strokes at acute stage?

2) Please clarify the outcome of interest is ‘satisfaction with care’ (not a general scale of life satisfaction)

3) Instrument to measure HRQoL and satisfaction with care could be mentioned briefly in the abstract

Method:

4) Whether those who died while in hospital and at 3-month follow-up were included in the analyses of mRS and HRQoL. Since the investigator already examined case fatality rates

5) Why both EQ-5D-5L and 15D were used to evaluate HRQoL, please specify?

6) Since the fact that “81% of the survivors were able and willing to respond to the questionnaires at follow-up, the response rate of PATSAT was 73%” and also HRQoL (73-75%), how to deal with missing data at 3-month follow-up

7) Is a sample of 200 cases powered enough to test the hypotheses? Why the number was chosen?

8) In the sentence stating with “In this last step we also considered the viability of the model and sometimes checked the effect of similar variables and also interaction terms”, it is unclear why similar variables were only sometimes checked, and what criteria for a selection. Please specify.

9) HRQoL data are frequently highly skewed, whether linear regression would work?

Results

10) Table 2: number of males, deaths at acute stage and follow-up could be presented as number and %. (fit in two columns as the variables represented earlier in the same table). A footnote can be used to specify the n(%) instead of mean (SD) where relevant.

11) Supplemental data: General patient satisfaction (item 32 of the PATSAT) was correlated (medium to high) with EQ5D (Table S5) and 15D, and inclusion of EQ5D in the model seems to be over adjustment (Table S8).

Discussion

12) Since data satisfaction with care (together with HRQoL) were collected at follow-up rather than at discharge, is there possibility of recall bias? If so, please acknowledge this as a limitation

13) It is acknowledgeable that the authors suggest that ‘improving home nursing possibilities and could respond to better, earlier rehabilitative efforts and increased social support of stroke survivors’. However, some discussions on more specific strategies would be more interested in for the readers.

Conclusion

14) In the sentence ‘Patient satisfaction was influenced negatively by stroke severity and positively by quality of life, and thrombolysis’, do you mean patient satisfaction was associated with better HRQoL and provision of intravenous thrombolysis’?

Minor edits

- Spell out uncommon acronyms such as TOAST (for ischaemic stroke subtype) and provide reference for the classification.

- ‘EQ-5D-5L’ in data collection instruments should be spelt out (i.e. EuroQOL-5 Dimension-5 levels)

- HRQoL can be used throughout the document instead of quality of life.

Reviewer #2: Comments to the Authors

General comments:

Thank you to the authors for providing me with the opportunity to review this interesting paper that investigates a subgroup of patients from the overall EuroHOPE study. Namely patients who have sustained an ischaemic stroke.

Overall, the study is a worthwhile addition to the literature and should be considered for publication, however, the flow of the manuscript needs to be improved.

My comments are relatively minor in nature (not seeking further analyses) and focus primarily on tightening the manuscript and some further contextualisation within the manuscript. For example I consider that the language pertaining to classification of the selected variables could be tightened (eg demographic, clinical, health economic). Additionally, population norms for the multi-attribute utility instruments should be included and considered in the broader context of the results.

Overall, the manuscript would also benefit from the review of an English language editor before resubmission. The manuscript would also benefit from a clearer explanation of the statistical methods and model specification.

Specific comments

ABSTRACT

Perhaps recast the Abstract in light of some of my comments.

INTRODUCTION

Line 48: please provide more contemporary references for 1 and 2. 1996 and 2010 not relevant when there is a burgeoning literature on the subject, particularly in the past 5 years.

Line 53: is this diverging socioeconomic background? Please be more specific. How many countries in Central Eastern Europe? – please contextualise this for the next sentence that states that Hungary was one of only 6 countries…

Line 56: perhaps say “with stroke being one of the only diseases that was modelled?’ You could also note the other major diseases that were modelled in this broader study to provide additional contextualisation.

Line 58: please include a comma To improve long-term outcome after stroke, it would be of major importance to identify social groups and features of post stroke care that are associated with fatality or worse HRQoL after stroke.

Please ask an English language Editor to tighten the manuscript – as noted in the general comments.

Line 61: is this diminished stroke outcome or outcomes? Also I understand that you mean there is an inverse relationship between stroke outcome and taxable income – perhaps you could tighten this sentence to describe the nature of this association.

Line 64: rather than say “we set forth”, please clearly state the aims of the paper in this last paragraph on this section and name these up as aims.

METHODS

Perhaps revise the first subheading to “Study setting and recruitment of participants”.

In light of the revised subheading, please swap the first paragraph with the second paragraph. The second paragraph about the recruitment of participants needs tightening. Perhaps say

Data presented here were collected within the framework of the overall EuroHOPE project [13] a longitudinal prospective study including patients with ischemic stroke (International Classification of Diseases I63). These patients were selected on the basis of their consecutive admission and the only other selection criteria applied was that this admission was a new-onset and their first ischemic stroke event. Additionally, this subgroup analysis of the broader EuroHOPE study adopted different inclusion criteria where persons with incomplete personal identification numbers, patients under 18 years of age, or with stroke admission during the previous 365 days, as well as patients with incomplete data retrospectively and/or follow-up for the period of 365 days were excluded from the broader study [19].

Line 88: perhaps say “or from the carer (as their legal representative), in case of patients whom were unable to consent” rather than “in case of patients unable to express themselves”.

Study design section

In regard to the “Study design” section and the parameters assessed in the study, I would suggest that this section needs tightening and additional clarification. After the ethics sentence the description of the selection and gathering of variables is somewhat confused. I would also suggest that the variables themselves would perhaps benefit from further classifications, and a rethinking of these classifications. For example, marital status should be separate from working status prior and after stroke. Marital status should be included in sociodemographic variables and working status could be classified within a health economics or economics classification. I like the idea of Table 1, however, further thought needs to be given to the subgroupings.

The study design section does not flow at all into the next section of data collection and instruments.

Data collection and instruments section

The first paragraph should describe the types of instruments that have been selected and why.

For example, please describe the EQ-5D-5L and 15D as multi-attribute utility instruments and what they measure – ie a health state utility. I’m not convinced that the authors understand what a health state utility is nor the reasons for these measurements, including an objective measure of quality of life for clinical assessment. Given that these instruments measure one of your key outcome measures, please provide a deeper understanding of these instruments and the concept of a health state utility in this section.

Similarly, please properly describe the summary scores of the other instruments described in this section, including the measurements and ranges of the scores and how the values should be interpreted.

Statistical analyses

This section is somewhat jumbled. Overall this section requires some re-ordering and tightening.

Perhaps break this section into two sections – one section that describes the model specification and another section that describes the statistical testing.

The model specification needs to commence from first principles and then describe the model used for each scenario. Then describe how the model was built and the testing scenarios.

Tests used for the continuous and categorical variables could be better described in the first paragraph. In additional it is not adequate to say that “the type of test depended on the measurement level of the variables”. Succinctly describe the reason for each test.

RESULTS

Line 174 - 183: you could label this section as “Participant characteristics”. Also please note here the number of patients that entered the study – namely N=200. Additionally, Table 2 is poorly ordered and typeset. For example, I would expect age and sex to be listed first. Also as described in the section above you have used the Mann-Whitney test for the age variable – I understand why this nonparametric test has been used – please describe this in your methods. You could also describe proportions for the categorical variables. Also, each Table needs to be stand alone – please describe the acronyms for each table in the Footnotes.

Line 190: change ‘social features’ to ‘sociodemographic variables’

Line 195 – 238: ‘Health-related quality of life’: The first paragraph could be labelled ‘Questionnaire completion’. In this subsection please provide a breakdown of each questionnaire. For example, was the completion rate for the generic multi-attribute utility instruments the same as the other instruments. Additionally, in the reporting of the summary statistics for each questionnaire in the associated table provide an n=x for the number of respondents whom completed.

In fact, in some sections of the paper (particularly the tables) I note that you have provided the proportions only of respondents– please also provide the numbers of respondents.

Please also compare the EQ-5D and 15 D results with population norms and minimal clinically important differences. Also draw this thread into the Discussion section of your paper.

DISCUSSION

Overall, please strengthen the first paragraph and section – that is provide a summary of the study, why it advances the literature and your key findings. Currently it provides key findings in a somewhat isolated manner.

Line 322 – overall the purpose of a multi-attribute utility instrument is to capture and assess complex physical and psychosocial health status/impact – particularly for people with complex and chronic disease if the correct instrument is chosen. To say that you need to avoid the physical health bias of HRQoL is therefore incorrect. Please understand what a utility value is measuring. If a more sensitive instrument was chosen, you could also assess the drivers for the utility value. The 15D will partially provide you with similar drivers – please investigate this in more detail. This will also complement your regression analyses.

Line 337: please say pre-stroke variables – not pre-stroke ‘ones’. Same with post-stroke.

Limitations; I would suggest that the selection of the EQ-5D could be a limitation. This instrument is known for its insensitivity for complex and chronic disease. I acknowledge that institutions such as NICE recommend the instrument – the prevalence of an instrument should not drive its selection. Please review the literature on this point and include some points in the Discussion.

Overall rather than saying worse quality of life you could also say reduced (from population norms too if you investigate and compare the utility values) and diminished from population norms. Also investigate minimal clinically important differences and consider how you can weave both population norms and MCID into the paper to provide further contextualisation to your Results and Discussion sections.

Once again, thank you to the authors and good luck with the next stages of the revisions.

6. PLOS authors have the option to publish the peer review history of their article (what does this mean?). If published, this will include your full peer review and any attached files.

Reviewer #1: Yes: Hoang Phan

Reviewer #2: No

---

## [Author Response · Author response to Decision Letter 0]

28 Aug 2020

Budapest, August 27, 2020

Seana Gall

Academic Editor

PLOS ONE

we submit the revised manuscript of Szőcs et al: Health related quality of life and satisfaction with care of stroke patients in Budapest: a substudy of the EuroHOPE project (Ms.No: PONE-D-20-05987).

We thank you for your useful comments and we especially thank the reviewers for taking the time to review our manuscript. We have considered each and every comment or recommendation and we have found them all to be for the improvement of the manuscript. We made the following changes to the manuscript:

A. To the comments of the Academic Editor

Question 1: Please ensure that your manuscript meets PLOS ONE's style requirements, including those for file naming. 

Answer 1: The manuscript has been revised to meet PLOS ONE’s style requirements, and the file naming has also been revised accordingly throughout the manuscript. 

Q 2: In your Methods section, please provide additional information about the participant recruitment method of the EuroHOPE project and the demographic details of your participants. Please ensure you have provided sufficient details to replicate the analyses such as: 

a) the recruitment date range (month and year) 

Answer: we have included the recruitment date range (September 2012 - May 2013) in the Methods section of the manuscript. According to Work Package 2 of the EuroHOPE project (HRQol protocol and patient satisfaction questionnaire – Stroke), patient inclusion in participating hospitals was planned to halt when n=200 was reached, according to the study protocol.

b) a description of any inclusion/exclusion criteria that were applied to participant recruitment 

Answer: we have included the selection criteria in the Methods section of the manuscript. “We prospectively included patients with ischemic stroke (International Classification Diseases, ICD-10, I63). According to the study protocol, inclusion of consecutive patients with first-ever ischemic stroke was planned up till including 200 patients. Exclusion criteria were history of previous ischemic stroke and refusal by the patient to participate in the survey. 

Patients were recruited between September 2012 and May 2013, and evaluated at the Department of Neurology of Semmelweis University, a stroke center with a catchment area covering districts 8 and 9 of Budapest, and the Ráckeve region.”

c) a table of relevant demographic details 

Answer: we have provided additional demographic details in Table 2 which was also revised according to the suggestions of both reviewers:

Table 2 Patient characteristics

Variable Variable level Overall Not District 8 District 8 p-value Response rate* (% of all 200)

N 200 159 41 

Age at stroke (mean (SD)) - 68.53 (12.86) 67.67 (12.53) 71.88 (13.72) 0.033 200 (100%)

Age at acute death (mean (SD)) - 75.00 (14.53) 78.70 (13.30) 67.60 (15.44) 0.198 15 (7.50%)

Age at follow-up death (mean (SD)) - 75.91 (12.28) 77.25 (11.66) 72.33 (13.90) 0.331 33 (16.50%)

Sex (%) Female 88 (44.0) 72 (45.3) 16 (39.0) 0.587 200 (100%)

 Male 112 (56.0) 87 (54.7) 25 (61.0) 

Years in education (mean (SD)) - 10.81 (2.80) 10.77 (2.71) 10.95 (3.14) 0.983 183 (91.50%)

Marital status (%) With partner 112 (56.0) 88 (55.3) 24 (58.5) 0.849 200 (100%)

 Living alone 88 (44.0) 71 (44.7) 17 (41.5) 

Pre-stroke employment (%) Not in employment 160 (81.6) 124 (79.5) 36 (90.0) 0.193 196 (98%)

 In employment 36 (18.4) 32 (20.5) 4 (10.0) 

Post-stroke employment (%) Not in employment 118 (88.1) 94 (87.0) 24 (92.3) 0.684 134 (67%)

 In employment 16 (11.9) 14 (13.0) 2 (7.7) 

Social type of pre-stroke dwelling (%) Home alone 64 (32.0) 54 (34.0) 10 (24.4) 0.189 200 (100%)

 Home with company 126 (63.0) 99 (62.3) 27 (65.9) 

 Institution 10 (5.0) 6 (3.8) 4 (9.8) 

Social type of post-stroke dwelling (%) Home alone 22 (16.5) 19 (17.4) 3 (12.5) 0.591 133 (66.50%)

 Home with company 96 (72.2) 79 (72.5) 17 (70.8) 

 Institution 15 (11.3) 11 (10.1) 4 (16.7) 

Acute fatality (%) No 185 (92.5) 149 (93.7) 36 (87.8) 0.343 200 (100%)

 Yes 15 (7.5) 10 (6.3) 5 (12.2) 

Three-month fatality (including acute death) (%) No 167 (83.5) 135 (84.9) 32 (78.0) 0.413 200 (100%)

 Yes 33 (16.5) 24 (15.1) 9 (22.0) 

NIHSS on admission (mean (SD)) - 7.88 (6.37) 7.36 (5.87) 9.88 (7.79) 0.141 200 (100%)

Discharge mRS (mean (SD)) (median (IQR)) - 2.68 (1.80) 2 (3) 2.57 (1.78) 2 (3) 3.10 (1.84) 3 (4) 0.111 200 (100%)

PATSAT score (mean (SD)) - 73.02 (22.38) 74.22 (22.35) 67.28 (22.17) 0.171 122 (61%)

EQ-5D utility index (mean (SD)) - 0.73 (0.29) 0.75 (0.28) 0.65 (0.34) 0.16 126 (63%)

15D utility index (mean (SD)) - 0.77 (0.16) 0.78 (0.16) 0.73 (0.17) 0.156 124 (62%)

Tests used: Mann-Whitney test for age variables, education, PATSAT, EQ-5D utility index, 15D utility index and NIHSS, chi-squared test for patient numbers and mRS.

* Response rate may mean data availably or number of events (i.e. death) depending on the variable. The number/percentage of respondents in complex scores and indices indicates the number of patients where these could be calculated. Individual items may have different response rate. In addition, 15D utility indices were calculated using imputation, thus the response rate of the final score may be higher than the one of the individual items.

NIHSS: National Institutes of Health Stroke Scale; mRS: modified Rankin Scale; PATSAT: EORTC IN-PATSAT32 questionnaire, assessing patient satisfaction; EQ-5D: the EuroQOL-5 Dimensions-5 Levels questionnaire; 15D: generic 15 dimensional measure of HRQoL (Sintonen 2001); IQR: interquartile range.

d) a statement as to whether your sample can be considered representative of a larger population 

Answer: we have included a statement: “Due to consecutive patient inclusion, our set of patients may be considered representative of ischemic stroke patients of this stroke center’s catchment area.”

e) a description of how participants were recruited 

Answer: we have included the description of how the participants were recruited in the Methods section of the manuscript.

“Ethical approval was obtained from the Regional and Institutional Committee of Science and Research Ethics of Semmelweis University (no. 98/2012) and written consent was obtained from each patient or from the carer in case of patients who were unable to consent. Initial evaluation of the patients was performed at admission by the physician on duty. The research investigators re-evaluated each case to ascertain that it is indeed an acute first-ever ischemic stroke.”

f) descriptions of where participants were recruited and where the research took place. Moreover, please provide more information on the catchment area and on the characteristics of the District 8.

Answer: We have added the following to the paper: “Patients were recruited between September 2012 and May 2013, and evaluated at the Department of Neurology of Semmelweis University, a stroke center with a catchment area covering Districts 8 and 9 of Budapest, and the Ráckeve region. The catchment area of this stroke center was defined by geographic availability, i.e. to be the nearest stroke center for the regions of the catchment area. From the socioeconomic point of view, District 8 ranks the last of the 23 districts of Budapest on the list of income; the other regions of the catchment area are in the medium or high income categories. Income constitutes the major socioeconomic difference between the regions in the catchment area. Due to consecutive patient inclusion, our set of patients may be considered representative of ischemic stroke patients of this stroke center’s catchment area.”

B. To the comments of Reviewer #1

Abstract

Q 1) ‘Higher severity in the acute phase’: do you mean more severe strokes at acute stage?

A 1: Yes, we have meant more severe strokes in the acute stage and we have corrected the text as such. 

Q 2) Please clarify the outcome of interest is ‘satisfaction with care’ (not a general scale of life satisfaction) 

A 2: we have specified the outcome of interest to be patient satisfaction with care and not a general scale of life satisfaction. We have checked and corrected this not only in the abstract, but also throughout the text of the manuscript. 

Q 3) Instrument to measure HRQoL and satisfaction with care could be mentioned briefly in the abstract

A 3: We have mentioned the instruments used to measure HRQoL and satisfaction with care in the abstract as well: 

“To evaluate HRQoL and patient satisfaction with care we used the EQ-5D-5L, 15D and EORTC IN PATSAT 32 questionnaires.”

Method:

Q 4) Whether those who died while in hospital and at 3-month follow-up were included in the analyses of mRS and HRQoL. Since the investigator already examined case fatality rates

A 4: We have collected HRQoL data only at follow-up. Therefore only data of survivors at follow-up have been used. While HRQoL can be defined for the deceased, we chose to create HRQoL-related models only for the survivors, as the imputation of values for deceased patients could have introduced a bias. As to mRS, we have included the results regarding the deceased, as well. We have made this clear in the manuscript by mentioning: 

“While HRQoL can be defined for the deceased, we chose to create HRQoL-related models only for the survivors, as the imputation of values for deceased patients could have introduced a bias.”

Q 5) Why both EQ-5D-5L and 15D were used to evaluate HRQoL, please specify?

A 5: we have complemented the Methods section by specifying this under the heading “Data collection instruments”:

“Stroke and its consequences have significant effects on health related quality of life (HRQoL). We applied indirect methods to evaluate HRQoL in our study with the use of multi-attribute utility (MAU) instruments: by the EuroQoL-5 Dimension-5 Levels (hereafter EQ-5D), developed by the EuroQoL group [Herdman et al, 2011], and by the HRQoL 15D instrument (hereafter 15D), [Sintonen 1994; Sintonen 2001a]. Both questionnaires are indirect generic instruments to evaluate HRQoL. EQ-5D addresses 5 dimensions: mobility, self-care, usual activities, pain/discomfort and anxiety/depression, grading these on 5 levels and a visual analog scale. The overall HRQoL described by the EQ-5D can be used to derive health state utility values, and has been found to be a valid descriptive system as a generic health outcome measure in patients with acute stroke [Golicki et al, 2015]. The 15D is a comprehensive, self-administered health-related QoL instrument, which consists of 15 dimensions: breathing, mental function, speech, vision, mobility, usual activity, vitality, hearing, eating, elimination, sleeping, distress, discomfort and symptoms, depression and sexual activity. While the EQ-5D is widely used [Vainiola 2010], the results might vary depending on the regression method used [Saarni 2006]. EQ-5D seems somewhat less sensitive when it comes to evaluating several chronic diseases [Vartianien 2017; Moock 2008, Kontodimoupoulos 2012, Heiskanen 2016]. The 15D performs better in terms of reliability, discrimination and responsiveness [Sintonen 2001b Arch Hell Med], and performs well after critical illness [Vainiola et al, 2010]. The agreement between the two utility measures was only moderate [Vainiola et al, 2010], and both the mean utilities and the standard deviation differs between EQ-5D and 15D [Richardson et al, 2015]. For these reasons – as recommended by Feeny et al [Feeny et al, 2019] – we decided to use multiple questionnaires in this study.”

Q 6) Since the fact that “81% of the survivors were able and willing to respond to the questionnaires at follow-up, the response rate of PATSAT was 73%” and also HRQoL (73-75%), how to deal with missing data at 3-month follow-up

A 6: We considered and added references to missing data both regarding respondent rate, and missing items within one patient. We have considered the rate of respondents acceptable as it was similar to the proportion reported in the literature, and added recent references to this statement:

 „We have considered the rate of respondents acceptable as it was similar or higher compared to the proportions reported after stroke [Mahesh 2018; Westerlind 2020]."

Mahesh PKB, Gunathunga MW, Jayasinghe S, Arnold SM, Liyanage SN. Factors influencing pre-stroke and post-stroke quality of life among stroke survivors in a lower middle-income country. Neurol Sci. 2018;39(2): 287-295. pmid:29103178

Westerlind E, Singh R, Persson HC, Sunnerhagen KS. Experienced pain after stroke: a cross-sectional 5-year follow-up study. BMC Neurol. 2020;20(1): 4. pmid:31910805”

The 15D missing data were imputed at the item level using an SPSS algorithm based on the age and gender of the patients (Sintonen 1994 and 2001). We have also addressed this in the paper:

“The 15D missing data were imputed at the item level using an SPSS algorithm for those with less than 4 missing items, based on the age and gender of the patients. [Sintonen 1994; Sintonen 2001]”.

Sintonen H. The 15D-measure of health-related quality of life. I. Reliability, validity and sensitivity of its health state descriptive system. National Centre for Health Program Evaluation, Working Paper 41, Melbourne 1994 https://www.monash.edu/__data/assets/pdf_file/0007/886633/wp41-1.pdf

Sintonen H. The 15D instrument of health-related quality of life: properties and applications. Ann Med. 2001;33(5): 328-336. pmid:11491191

Q 7) Is a sample of 200 cases powered enough to test the hypotheses? Why the number was chosen?

A 7: According to Work Package 2 of the EuroHOPE project, patient inclusion in participating hospitals was planned to halt when n=200 was reached, according to the study protocol. That was intended to be an exploratory study with no formal sample size estimation. We added to the manuscript the following: 

“The present study was conducted according to Work Package 2 of the EuroHOPE project (“HRQol protocol and patient satisfaction questionnaire – Stroke”). We prospectively included patients with ischemic stroke (International Classification Diseases, ICD-10, I63). According to the study protocol, inclusion of consecutive patients with first-ever ischemic stroke was planned up till including 200 patients. That was intended to be an exploratory study with no formal sample size estimation. Exclusion criteria were history of previous ischemic stroke and refusal by the patient to participate in the survey.” 

Q 8) In the sentence stating with “In this last step we also considered the viability of the model and sometimes checked the effect of similar variables and also interaction terms”, it is unclear why similar variables were only sometimes checked, and what criteria for a selection. Please specify.

A 8: The sentence was indeed poorly worded. We have corrected it. The similar variables in this context meant the pre-stroke and post-stroke version of the same variables (e.g. working status) and the original and discretized (pooled) version of the same variables (e.g. original mRS with categories 1-5 or discretized version with 0-1 and 2-5 pooled). All of these for all variables were considered during model building. The corrected part in the paper reads the following: “

In this last step we also considered the viability of the model and checked the effect of pre-stroke and post-stroke version and the original and pooled version of the same variables and also interaction terms. We also aimed to synchronize models with similar dependent variables.”

Q 9) HRQoL data are frequently highly skewed, whether linear regression would work?

A 9: Goodness of fit was tested using statistical and visual tools. Shapiro-Wilk test and quantile-quantile plot were used for linear regression. The only model that showed significant (Shapiro-Wilk p=0.009) fitting problems among the HRQoL models was the EQ-5D multiple linear regression model including only pre-stroke variables (Table S1). Nonetheless, we have addressed this fitting problem in the manuscript: 

“The only model that showed significant fitting problems (Shapiro-Wilk p=0.009) among the HRQoL models was the EQ-5D multiple linear regression model excluding post-stroke variables (Table S1). The result of the quantile-quantile plot was more dubious. As this was just a sub-analysis and the variable effects are similar in the other models this does not undermine our overall conclusions.”

Results

Q 10) Table 2: number of males, deaths at acute stage and follow-up could be presented as number and %. (fit in two columns as the variables represented earlier in the same table). A footnote can be used to specify the n(%) instead of mean (SD) where relevant.

A 10: We have added the percentages to the table and overall revised it according to the suggestions of both reviewers.

Q 11) Supplemental data: General patient satisfaction (item 32 of the PATSAT) was correlated (medium to high) with EQ5D (Table S5) and 15D, and inclusion of EQ5D in the model seems to be over adjustment (Table S8).

A 11: In our revised manuscript we present the association between the sub-item of general patient satisfaction (item 32 of the PATSAT questionnaire) and quality of life indices in Table S5. We thought it would be beneficial to see the effect of EQ5D on PATSAT while also considering the effect of other variables. In the revised manuscript Table S3 (all tables have been revised, thus numbers changed) also includes the models, where EQ-5D and mRS are not included.

Discussion

Q 12) Since data satisfaction with care (together with HRQoL) were collected at follow-up rather than at discharge, is there possibility of recall bias? If so, please acknowledge this as a limitation

A 12: At follow-up we have tried to contact each and every patient and carer using the phone numbers and post addresses given during the hospital stay for the index stroke. It is possible that among those who did not come in for an in-person check-up, the rate of the less satisfied patients was higher. This is indeed a limitation of the study as we have acknowledged. Further, as 3 months elapsed from the index stroke when patient satisfaction was evaluated, recall bias might be possible – we also mention this among the limitation of the study:

“Fourth, among those who did not return for an in-person check-up for the follow-up the rate of less satisfied patients may have been higher, and as the survey was done at 3 months after stroke, recall bias may have had an effect on patient satisfaction.”

Q 13) It is acknowledgeable that the authors suggest that ‘improving home nursing possibilities and could respond to better, earlier rehabilitative efforts and increased social support of stroke survivors’. However, some discussions on more specific strategies would be more interested in for the readers. 

A 13: We have completed the Discussions section mentioning more specific strategies that could be applied in frame of home nursing. 

“Our results suggest that shorter hospital stays followed by well-organized regular outpatient care could contribute to better post-stroke HRQoL. These outpatient interventions could be motor or non-motor rehabilitation sessions, regular home-based check-ups for improving the adherence to secondary prevention, nurses available for administering drugs and assisting the patients in their every-day grooming and medical attendance.” 

Conclusion

Q 14) In the sentence ‘Patient satisfaction was influenced negatively by stroke severity and positively by quality of life, and thrombolysis’, do you mean patient satisfaction was associated with better HRQoL and provision of intravenous thrombolysis’?

A 14: Yes, this is what we have meant by the quoted sentence. We reworded this statement as suggested, and moved this statement to the Discussion section: 

“Patient satisfaction was already known to be one of the most important predictors of HRQoL [Nunes 2017]. In addition to HRQoL, stroke severity also has a major impact on patient satisfaction: more severe strokes are associated with lower global satisfaction. On the other hand, the provision of thrombolysis had a significant positive impact on global patient satisfaction and the subcategory of satisfaction with nurses and care organization. We have not found any description of this association yet.”

Minor edits 

- Spell out uncommon acronyms such as TOAST (for ischaemic stroke subtype) and provide reference for the classification. 

A: we have spelled out the acronym and provided the reference in the Data collection instruments chapter of the Methods section and also in the legend of Table 1.

- ‘EQ-5D-5L’ in data collection instruments should be spelt out (i.e. EuroQOL-5 Dimension-5 levels) 

A: we have spelled out EQ-5D-5L as the reviewer has suggested. 

- HRQoL can be used throughout the document instead of quality of life.

A: we have corrected the text and have used HRQoL instead of quality of life throughout the text of the manuscript. Also, we changed the term in the title of the manuscript from “Quality of life..” to “Health related quality of life…”

C. To the comments of Reviewer #2

Q1: General comments:

Thank you to the authors for providing me with the opportunity to review this interesting paper that investigates a subgroup of patients from the overall EuroHOPE study. Namely patients who have sustained an ischaemic stroke.

Overall, the study is a worthwhile addition to the literature and should be considered for publication, however, the flow of the manuscript needs to be improved.

My comments are relatively minor in nature (not seeking further analyses) and focus primarily on tightening the manuscript and some further contextualisation within the manuscript. For example I consider that the language pertaining to classification of the selected variables could be tightened (eg demographic, clinical, health economic). Additionally, population norms for the multi-attribute utility instruments should be included and considered in the broader context of the results.

Overall, the manuscript would also benefit from the review of an English language editor before resubmission. The manuscript would also benefit from a clearer explanation of the statistical methods and model specification.

A1: We thank you for taking the time to review the manuscript. We are grateful for your work and we highly appreciate your judgement. We have tried to follow each of your recommendations and thoroughly revised our manuscript. We have modified the terms and reclassified the selected variables to sociodemographic, socioeconomic, clinical features and outcome measures in the revised Table 1. 

To improve the flow of the manuscript, we have rearranged the text and the order of the tables. In the new set of revised tables:

Table 1 introduces the parameters assessed in the study

Table 2 presents patient characteristics,

Table 3 summarizes the results of regression models of acute and late case fatality

Tables 4-6 present the predictors of EQ-5D and 15D utility indices and of patient satisfaction.

We present the results of additional analyses as supplementary tables:

S1 Table Predictors of EQ-5D utility index in multiple linear regression model excluding post-stroke variables

S2 Table Predictors of 15D utility index in multiple linear regression model excluding post-stroke variables

S3 Table Predictors of PATSAT score and its subcategories in multiple linear regression and ordered logistic regression models excluding post-stroke variables

S4 Table Correlation between EQ-5D utility index, 15D utility index, PATSAT score and its subcategories

S5 Table Association between the sub-item of general patient satisfaction (item 32 of the PATSAT questionnaire) and quality of life indices.

We have included population norms for Hungary of the multi-attribute utility instrument EQ-5D, and mentioned that no such values are available for 15D for the Hungarian population. 

We have asked an English language editor to review the text before resubmission to improve the manuscript.

Our statistician has rewritten the description of the statistical methods in order to provide a clearer explanation of the statistical methods and model specification.

Specific comments 

Q2: ABSTRACT

Perhaps recast the Abstract in light of some of my comments. 

We have rewritten the Abstract of the manuscript.

“Background: Disadvantaged socioeconomic status is associated with higher stroke incidence and mortality, and higher readmission rate. We aimed to assess the effect of socioeconomic factors on case fatality, health related quality of life (HRQoL), and satisfaction with care of stroke survivors in the framework of the European Health Care Outcomes, Performance and Efficiency (EuroHOPE) study in Hungary, one of the leading countries regarding stroke mortality. 

Methods: We evaluated 200 consecutive patients admitted for first-ever ischemic stroke in a single center and performed a follow-up at 3 months after stroke. We recorded pre- and post-stroke socioeconomic factors, and assessed case fatality, HRQoL and patient satisfaction with the care received. Stroke severity at onset was scored by the National Institutes of Health Stroke scale (NIHSS), disability at discharge from acute care was evaluated by the modified Rankin Score (mRS). To evaluate HRQoL and patient satisfaction with care we used the EQ-5D-5L, 15D and EORTC IN PATSAT 32 questionnaires. 

Results: At 3 months after stroke the odds of death was significantly increased by stroke severity (NIHSS, OR=1.209, 95%CI: 1.125-1.299, p<0.001) and age (OR=1.045, 95%CI: 1.003-1.089, p=0.038). In a multiple linear regression model, independent predictors of HRQoL were age, disability at discharge, satisfaction with care, type of social dwelling after stroke, length of acute hospital stay and rehospitalization. Satisfaction with care was influenced negatively by stroke severity (Coef. =-1.111, 95%C.I.: -2.159- -0.062, p=0.040), and positively by having had thrombolysis (Coef. =25.635, 95%C.I.: 5.212-46.058, p=0.016) and better HRQoL (Coef.=22.858, 95%C.I.: 6.007-39.708, p=0.009).

Conclusion: In addition to age, disability, and satisfaction with care, length of hospital stay and type of social dwelling after stroke also predicted HRQoL. Long-term outcome after stroke could be improved by reducing time spent in hospital, i.e. by developing home care rehabilitation facilities thus reducing the need for readmission to inpatient care.”

INTRODUCTION

Line 48: please provide more contemporary references for 1 and 2. 1996 and 2010 not relevant when there is a burgeoning literature on the subject, particularly in the past 5 years.

A: we have provided contemporary references instead of those given previously:

“Wang R, Langhammer B. Predictors of quality of life for chronic stroke survivors in relation to cultural differences: A literature review. Scand J Caring Sci. 2018;32(2): 502-514. pmid:28949412

Ramos-Lima MJM, Carvalho Brasileiro I, Lima TL, Braga-Neto P. Quality of life after stroke: impact of clinical and sociodemographic factors. Clinics (Sao Paulo). 2018;73: e418. pmid:30304300.”

Q3: Line 53: is this diverging socioeconomic background? Please be more specific. How many countries in Central Eastern Europe? – please contextualise this for the next sentence that states that Hungary was one of only 6 countries…

A3: We have focused the text, and have presented the context for introducing the EuroHOPE study: 

“The impact of differences in social background on post-stroke health-related quality of life (HRQoL) has not been sufficiently evaluated in Central-Eastern European countries. Although there are some data on the consequences of social inequalities on stroke features among neighborhoods within the capital city of Hungary [Szőcs et al 2019] reports evaluating HRQoL data after stroke are scarce from Central-Eastern European countries [Prevolnik et al, 2019]. 

The European Health Care Outcomes, Performance and Efficiency (EuroHOPE) project compared health system performance in 6 European countries (Italy, Finland, Sweden, Scotland, the Netherlands and Hungary) [Hakkinen et al, 2013], evaluating stroke, acute myocardial infarction, hip fracture, breast cancer, and low-birth-weight. In the EuroHOPE study Hungary had the highest stroke incidence, the largest all-cause case fatality 1 year after stroke and the largest regional differences [Malmivaara 2015; EUROHOPE Atlas website, 2016].”

Q4: Line 56: perhaps say “with stroke being one of the only diseases that was modelled?’ You could also note the other major diseases that were modelled in this broader study to provide additional contextualisation.

A4: We have tightened the text, and made clear the countries involved in the EuroHOPE study, and that stroke was one of 5 health conditions, presenting the 4 other analyzed conditions as well: 

“The European Health Care Outcomes, Performance and Efficiency (EuroHOPE) project compared health system performance in 6 European countries (Italy, Finland, Sweden, Scotland, the Netherlands and Hungary) [Hakkinen et al, 2013], evaluating stroke, acute myocardial infarction, hip fracture, breast cancer, and low-birth-weight.”

We also made it clear in the Methods section that our study has been part of the second phase of the EuroHOPE study, i.e. not a retrospecive analysis of healthcare administrative data done in the first phase of EuroHOPE, but a prospective follow-up study of individual patients:

“Data presented here were collected within the framework of the EuroHOPE project [Hakkinen 2013], a retrospective study based on national hospital healthcare administrative records in 6 European countries, with the index ischemic stroke between 2006-2008 [Malmivaara 2015, Malmivaara 2013]. The present study was conducted according to Work Package 2 of the EuroHOPE project (“HRQol protocol and patient satisfaction questionnaire – Stroke”). We prospectively included patients with ischemic stroke (International Classification Diseases, ICD-10, I63). According to the study protocol, inclusion of consecutive patients with first-ever ischemic stroke was planned up till including 200 patients.”

Q5: Line 58: please include a comma To improve long-term outcome after stroke, it would be of major importance to identify social groups and features of post stroke care that are associated with fatality or worse HRQoL after stroke. 

A5: we have corrected the text by including the comma. 

Q6:Please ask an English language Editor to tighten the manuscript – as noted in the general comments.

A6: we are sorry for taking your time with English language mistakes during the reviewing process. We have asked for the help of an English language editor during manuscript revision. 

Q6: Line 61: is this diminished stroke outcome or outcomes? Also I understand that you mean there is an inverse relationship between stroke outcome and taxable income – perhaps you could tighten this sentence to describe the nature of this association. 

A6: Indeed, we mean that there is an inverse relationship between stroke outcome and taxable income in the 23 neighborhoods of Budapest, and there is also an inverse relationship between the mean annual taxable income and the age of stroke onset. We have made the paragraph more compact:

“We previously found worse stroke outcome in patients residing in the poorest district of Budapest compared to those of the wealthiest neighborhood [Szőcs 2012; Folyovich 2015], and also found that the lower the annual taxable income, the lower the age at stroke onset in the 23 districts of Budapest [Szőcs 2019].”

Q7: Line 64: rather than say “we set forth”, please clearly state the aims of the paper in this last paragraph on this section and name these up as aims. 

A7: we have focused and rephrased the aims of the study as the Reviewer has recommended: 

“The aim of the present study was to assess the impact of stroke-related, demographic and socioeconomic factors on acute and 3-months case fatality, on HRQoL and on satisfaction with care of patients after stroke.” 

Q8: METHODS

Perhaps revise the first subheading to “Study setting and recruitment of participants”.

A8: AS recommended, we have changed the subheading to “Study setting and recruitment of participants” 

Q9: In light of the revised subheading, please swap the first paragraph with the second paragraph. The second paragraph about the recruitment of participants needs tightening. Perhaps say 

Data presented here were collected within the framework of the overall EuroHOPE project [13] a longitudinal prospective study including patients with ischemic stroke (International Classification of Diseases I63). These patients were selected on the basis of their consecutive admission and the only other selection criteria applied was that this admission was a new-onset and their first ischemic stroke event. Additionally, this subgroup analysis of the broader EuroHOPE study adopted different inclusion criteria where persons with incomplete personal identification numbers, patients under 18 years of age, or with stroke admission during the previous 365 days, as well as patients with incomplete data retrospectively and/or follow-up for the period of 365 days were excluded from the broader study [19].

A9: we have swapped the second paragraph with the first one. We have also rewritten this paragraph as you suggested, combining it with the response and recommendations of the other Reviewer:

“Data presented here were collected within the framework of the EuroHOPE project [Hakkinen 2013], a retrospective study based on national hospital healthcare administrative records in 6 European countries, with the index ischemic stroke between 2006-2008 [Malmivaara 2015, Malmivaara 2013]. The present study was conducted according to Work Package 2 of the EuroHOPE project (“HRQol protocol and patient satisfaction questionnaire – Stroke”). We prospectively included patients with ischemic stroke (International Classification Diseases, ICD-10, I63). According to the study protocol, inclusion of consecutive patients with first-ever ischemic stroke was planned up till including 200 patients. That was intended to be an exploratory study with no formal sample size estimation. Exclusion criteria were history of previous ischemic stroke and refusal by the patient to participate in the survey.”

Q10: Line 88: perhaps say “or from the carer (as their legal representative), in case of patients whom were unable to consent” rather than “in case of patients unable to express themselves”.

A10: we have modified the text following your recommendation. 

Q11: Study design section 

In regard to the “Study design” section and the parameters assessed in the study, I would suggest that this section needs tightening and additional clarification. After the ethics sentence the description of the selection and gathering of variables is somewhat confused. I would also suggest that the variables themselves would perhaps benefit from further classifications, and a rethinking of these classifications. For example, marital status should be separate from working status prior and after stroke. Marital status should be included in sociodemographic variables and working status could be classified within a health economics or economics classification. I like the idea of Table 1, however, further thought needs to be given to the subgroupings. 

A11: Corresponding to your recommendation, and also in an attempt to satisfy the requests of the other reviewer, we have rewritten and considerably tightened this section, making it clearer:

“Ethical approval was obtained from the Regional and Institutional Committee of Science and Research Ethics of Semmelweis University (no. 98/2012) and written consent was obtained from each patient or from the carer in case of patients who were unable to consent. Initial evaluation of the patients was performed at admission by the physician on duty. The research investigators re-evaluated each case to ascertain that it is indeed an acute first-ever ischemic stroke. As the next step, clinical features, pre-stroke demographic and socioeconomic factors (Table 1.) were recorded by a standard questionnaire. Follow-up was performed three months after the onset of stroke by personal visit or by phone interview to assess post-stroke socioeconomic factors, HRQoL and patient satisfaction with care. Length of post-acute hospital stay (i.e. institutional rehabilitation or rehospitalization for any other reason after discharge from the acute setting) was recorded. Post-acute hospital stay was defined as the number of all inpatient days after discharge from the index hospital admission within the three months of follow-up. 

Outcome measures were case fatality during the initial hospitalization and during follow-up, HRQoL and satisfaction with care of the survivors at follow-up.” 

According to your suggestion, we have restructured Table 1:

Table 1. Parameters assessed in the study

Sociodemographic factors • age

• gender

• education

• marital status prior and after stroke 

Socioeconomic factors • working status prior and after stroke

• dwelling prior to stroke: 

o location (resident of poorest District 8 or other neighborhood) 

o conditions (home or institute) 

o type of social dwelling (alone, with company, with family help, with professional help)

• dwelling after stroke: 

o conditions 

o type of social dwelling 

Clinical features • pre-stroke clinical factors: degree of assistance needed, pre-stroke depression or dementia, vascular risk factors, prior drug therapy

• strictly stroke related: severity at onset in NIHSS, disability at discharge from acute care in mRS, stroke type (TOAST classification)

• management related: thrombolysis status, necessity of intensive care or endarterectomy, motor rehabilitation therapy, post-stroke drug therapy, acute and post-acute length of stay (LOS)

Outcome measures • case fatality: acute (inpatient) and at follow-up

• health-related quality of life of survivors at follow-up

• patient satisfaction with care at follow-up

NIHSS: National Institutes of Health Stroke Scale; mRS: modified Rankin Scale; TOAST: Trial of Org 10172 in Acute Stroke Treatment; LOS: length of stay.

Q12: The study design section does not flow at all into the next section of data collection and instruments.

A12: we have thoroughly checked and extensively rewritten both sections (see above and below) so that the study design section flows into the next section of data collection and instruments.

Q13: Data collection and instruments section 

The first paragraph should describe the types of instruments that have been selected and why.

For example, please describe the EQ-5D-5L and 15D as multi-attribute utility instruments and what they measure – ie a health state utility. I’m not convinced that the authors understand what a health state utility is nor the reasons for these measurements, including an objective measure of quality of life for clinical assessment. Given that these instruments measure one of your key outcome measures, please provide a deeper understanding of these instruments and the concept of a health state utility in this section. 

A13: We agree with our Reviewer that we had been superficial in this section. Therefore we have completely rewritten the relevant section of the paper in the following way:

“Stroke and its consequences have significant effects on health related quality of life (HRQoL). We applied indirect methods to evaluate HRQoL in our study with the use of multi-attribute utility (MAU) instruments: by the EuroQoL-5 Dimension-5 Levels (hereafter EQ-5D), developed by the EuroQoL group [Herdman et al, 2011], and by the HRQoL 15D instrument (hereafter 15D), [Sintonen 1994; Sintonen 2001a]. Both questionnaires are indirect generic instruments to evaluate HRQoL. EQ-5D addresses 5 dimensions: mobility, self-care, usual activities, pain/discomfort and anxiety/depression, grading these on 5 levels and a visual analog scale. The overall HRQoL described by the EQ-5D can be used to derive health state utility values, and has been found to be a valid descriptive system as a generic health outcome measure in patients with acute stroke [Golicki et al, 2015]. The 15D is a comprehensive, self-administered health-related QoL instrument, which consists of 15 dimensions: breathing, mental function, speech, vision, mobility, usual activity, vitality, hearing, eating, elimination, sleeping, distress, discomfort and symptoms, depression and sexual activity. While the EQ-5D is widely used [Vainiola 2010], the results might vary depending on the regression method used [Saarni 2006]. EQ-5D seems somewhat less sensitive when it comes to evaluating several chronic diseases [Vartianien 2017; Moock 2008, Kontodimoupoulos 2012, Heiskanen 2016]. The 15D performs better in terms of reliability, discrimination and responsiveness [Sintonen 2001b Arch Hell Med], and performs well after critical illness [Vainiola et al, 2010]. The agreement between the two utility measures was only moderate [Vainiola et al, 2010], and both the mean utilities and the standard deviation differs between EQ-5D and 15D [Richardson et al, 2015]. For these reasons – as recommended by Feeny et al [Feeny et al, 2019] – we decided to use multiple questionnaires in this study.”

Q14: Similarly, please properly describe the summary scores of the other instruments described in this section, including the measurements and ranges of the scores and how the values should be interpreted. 

A14: We have described the summary scores of the other instruments used, including the measurement and ranges of the scores and how the values should be interpreted. 

For the stroke and disability measures:

“Initial stroke severity was assessed by the National Institutes of Health Stroke Scale (NIHSS), [Brott 1989] on admission, whereas disability at discharge was scored by the modified Rankin Scale (mRS), [Rankin, 1957]. The NIHSS is a 15-item neurological examination scale used to evaluate the effect of stroke on the level of consciousness, language, neglect, visual-field loss, extraocular movement, motor strength, coordination, dysarthria, and sensory loss. The NIHSS score ranges from 0 to 42, higher scores representing more severe states. The mRS addresses disability ranging from 0 (asymptomatic) to 6 (death). The classification of ischemic stroke was performed using Trial of Org 10172 in Acute Stroke Treatment (TOAST) criteria [Adams 1999]. According to TOAST, ischemic stroke can be classified into large-artery atherosclerosis, cardioembolism, small-vessel disease, stroke of other determined etiology, and stroke of undetermined etiology.” 

For the evaluation of patient satisfaction:

“We have also assessed the experience of patients with the care received during the hospital stay by EORTC IN-PATSAT32 (hereafter PATSAT), a tool that was developed by the European Organization for Research and Treatment of Cancer [EORTC 2005]. It evaluates 32 items regarding the satisfaction with technical competence, information provision, interpersonal skills, availability, waiting time, access, comfort and overall care perception. The following terms can be used: global patient satisfaction expressed by the PATSAT score of the questionnaire; three subcategories of patient satisfaction (with doctors, with nurses and with services and care organization) and the item number 32 of the questionnaire which is the general satisfaction with care received during the hospital stay of the patient. A higher scale score represents a higher level of satisfaction with care.”

Q15:Statistical analyses 

This section is somewhat jumbled. Overall this section requires some re-ordering and tightening.

Perhaps break this section into two sections – one section that describes the model specification and another section that describes the statistical testing.

The model specification needs to commence from first principles and then describe the model used for each scenario. Then describe how the model was built and the testing scenarios.

Tests used for the continuous and categorical variables could be better described in the first paragraph. In additional it is not adequate to say that “the type of test depended on the measurement level of the variables”. Succinctly describe the reason for each test.

A15: We thank the reviewer for the detailed help with the better organization of this section. We have completely revised and shortened this section according to the suggestions of our Reviewer.

Q16:RESULTS

Line 174 - 183: you could label this section as “Participant characteristics”. Also please note here the number of patients that entered the study – namely N=200. Additionally, Table 2 is poorly ordered and typeset. For example, I would expect age and sex to be listed first.

A16: We have completely reorganized Table 2 according to the suggestions of both Reviewers, and in the revised table we present patient features as suggested. 

Q17: Also as described in the section above you have used the Mann-Whitney test for the age variable – I understand why this nonparametric test has been used – please describe this in your methods.

A17: We have corrected the Statistical tests subsection, as required by our Reviewer.

Q18: You could also describe proportions for the categorical variables.

A18: We have added proportions to Table 2.

Q19: Also, each Table needs to be stand alone – please describe the acronyms for each table in the Footnotes. 

A19: We have described the acronyms for each table in the Footnotes.

Q20:Line 190: change ‘social features’ to ‘sociodemographic variables’

A20: We have changed ‘social features’ to ‘sociodemographic variables’ or ‘sociodemographic factors’ throughout the text of the manuscript. 

Q21: Line 195 – 238: ‘Health-related quality of life’: The first paragraph could be labelled ‘Questionnaire completion’. In this subsection please provide a breakdown of each questionnaire. For example, was the completion rate for the generic multi-attribute utility instruments the same as the other instruments. Additionally, in the reporting of the summary statistics for each questionnaire in the associated table provide an n=x for the number of respondents whom completed. 

A21: Thank you for the suggestion, we have changed the paragraph in question, and Table 2 accordingly. In the text:

“Questionnaire completion

Assessment of HRQoL and patient satisfaction at follow-up could have been possible in 81% of the survivors (136 cases alive, able and willing to express themselves, 7 unable - aphasic, demented or comatose, 24 patients declined to answer). The response rates among the 167 patients surviving 3 months were 122 (73%) for PATSAT score, 126 (75%) for EQ-5D utility index and 124 (74%) for 15D utility index. The difference from 136 is due to incomplete answers at item level. The response rate for HRQoL and patient satisfaction questionnaires were almost the same (73%-75%), while initial stroke severity and disability data (i.e. NIHSS and mRs) could be gathered from all 167 survivors. The 15D missing data were imputed at the item level using an SPSS algorithm for those with less than 4 missing items, based on the age and gender of the patients[Sintonen 1994; Sintonen 2001a]”.

Q22:In fact, in some sections of the paper (particularly the tables) I note that you have provided the proportions only of respondents– please also provide the numbers of respondents.

A22: We have provided the number of respondents in Table 2 and in the text.

Q23:Please also compare the EQ-5D and 15 D results with population norms and minimal clinically important differences. Also draw this thread into the Discussion section of your paper.

A23: In the revised manuscript we added two paragraphs on the issues brought up by our Reviewer: the first is on utility indexes for stroke patients both for EQ-5D and 15D. The second paragraph is on minimally important differences for utility indices. We could compare our results to Hungarian population norms for EQ-5D based on published Hungarian EQ-5D population norms (Janssen and Szende, 2104). For 15D we could not find population norms for Hungary. The 2 paragraphs read as follows:

“Although dwellers of the socioeconomically disadvantaged District 8 scored less by both health utility measures compared to residents of wealthier regions (Table 1), similarly to case fatality, the number of patients was not sufficient for this difference to reach statistical significance. We have considered the rate of respondents acceptable as it was similar or higher compared to the proportions reported after stroke [Mahesh 2018; Westerlind 2020]. Nonetheless we found that the average EQ-5D index in both subgroups (0.65±0.34 for District 8 and 0.75±0.28 for other regions) are below the total population norm of 0.82 [Janssen and Szende, 2014]. Using EQ-5D the health state utility values after stroke were relatively stable in time: in a systematic review of a total of 199 publications on stroke the median value was 0.63 in studies before 2013 versus 0.65 in studies after 2013 [Betts et al 2020]. For 15D, the difference in health utility values between the socioeconomically disadvantaged district and the other areas also did not reach the level of statistical significance in our study (0.73±0.17 for District 8 and 0.78±0.16 for other regions). These values for 15D are within the range reported for stroke patients with or without vision problems (0.73 – 0.89; [Sand et al, 2015]). It has been reported that EQ-5D and 15D should not be used interchangeably in economic evaluations after stroke, as the utility scores generated from the two instruments, although correlated well, they differed significantly from each other [Lunde 2013]. Using the same health state utility measures 3 months after intracerebral hemorrhage, predictors for lower HRQoL by both scales were higher NIHSS and older age, with similar ORs for EQ-5D-5L, and 15D [Sallinen 2019]. 

Minimally important differences (MIDs) for health state utilities vary by measure and methods and are not well established [Mouelhi et al, 2020]. It has been suggested that for EQ-5D differences among health state utilities of at least 0.036 can be considered clinically important [Feeny 2005]. For EQ-5D, the mean MID among non-stroke patients was 0.074 (range -0.011–0.140) [Walters and Braizer, 2005]. In a Korean version of the EQ-5D-3L questionnaire, MID in stroke patients ranged from 0.08 to 0.12 [Kim 2015], and a similar value of 0.10 was reported by Chen et al [Chen et al 2016]. Considering the population norms of EQ-5D reported for Hungary (0.82, [Janssen and Szende 2014]) the mean state utility value for EQ-5D in our patients (0.65 for District 8 and 0.75 for the wealthier regions) is considerably smaller, i.e. HRQoL is obviously adversely affected in stroke survivors 3 months after the acute event with a signal for worse HRQoL in District 8. As far as 15D is concerned, there is a lack of information on MID values in stroke patients, and also no population norms are known for Hungary. In other patient groups MID for 15D was estimated between 0.01 – 0.03 [Alanne et al, 2015]. Although by standard statistical methods – similarly to our findings for EQ-5D – the difference between health state utilities by the 15D was not statistically significant, the difference between the mean values of DDistrict 8 and other regions (i.e. 0.78 – 0.73 = 0.05) suggests that HRQoL after stroke is worse in socioeconomically deprived regions than in wealthier areas.”

Q24: DISCUSSION

Overall, please strengthen the first paragraph and section – that is provide a summary of the study, why it advances the literature and your key findings. Currently it provides key findings in a somewhat isolated manner.

A24: According to the suggestion of our Reviewer, we have rephrased the first paragraph of the discussion as follows:

“In a consecutive sample of 200 patients with acute ischemic stroke we evaluated predictors of case fatality, health-related quality of life and patient satisfaction. Predictors of acute in-hospital case fatality were stroke severity and living alone prior to stroke. At 3 months after stroke, similarly to previous reports, we also found that case fatality related to initial stroke severity and age. Health utility index was lower after stroke than the population norm for EQ-5D for Hungary, and utility indices tended to be lower in the disadvantaged District 8 than in wealthier regions both by EQ-5D and 15D, suggesting that in addition to the consequences of stroke, the general living standard of the patient also effects post-stroke HRQoL. Independent predictors of HRQoL at 3 months after stroke, similarly to other reports, were age, disability at discharge from the acute hospitalization, stroke type, and patient satisfaction. As far as we know, it has not been reported previously that acute and post-acute length of hospital stay after stroke also affect HRQoL. We found evidence that post-acute inpatient management might adversely affect HRQoL: longer than 5 days post-acute LOS was associated with poorer HRQoL. In our study the single independent predictor of all subcategories of patient satisfaction was HRQoL. Global patient satisfaction was influenced negatively by initial stroke severity and positively by thrombolysis and HRQoL. According to our knowledge, this positive effect of the provision of thrombolysis on patient satisfaction has not been described before.”

Q25: Line 322 – overall the purpose of a multi-attribute utility instrument is to capture and assess complex physical and psychosocial health status/impact – particularly for people with complex and chronic disease if the correct instrument is chosen. To say that you need to avoid the physical health bias of HRQoL is therefore incorrect. Please understand what a utility value is measuring. If a more sensitive instrument was chosen, you could also assess the drivers for the utility value. The 15D will partially provide you with similar drivers – please investigate this in more detail. This will also complement your regression analyses.

A25: We agree with our Reviewer that the physical health status is also critically important in health-related quality of life, therefore deleted the wrong sentence. We discussed the effect of stroke on HRQoL evaluated by the two multi-attribute utility instruments used in our study as well, and reviewed the state-of-art including several recent references to our manuscript:

“We have considered the rate of respondents acceptable as it was similar or higher compared to the proportions reported after stroke [Mahesh 2018; Westerlind 2020]. Nonetheless we found that the average EQ-5D index in both subgroups (0.65±0.34 for District 8 and 0.75±0.28 for other regions) are below the total population norm of 0.82 [Janssen and Szende, 2014]. Using EQ-5D the health state utility values after stroke were relatively stable in time: in a systematic review of a total of 199 publications on stroke the median value was 0.63 in studies before 2013 versus 0.65 in studies after 2013 [Betts et al 2020]. For 15D, the difference in health utility values between the socioeconomically disadvantaged district and the other areas also did not reach the level of statistical significance in our study (0.73±0.17 for District 8 and 0.78±0.16 for other regions). These values for 15D are within the range reported for stroke patients with or without vision problems (0.73 – 0.89; [Sand et al, 2015]). It has been reported that EQ-5D and 15D should not be used interchangeably in economic evaluations after stroke, as the utility scores generated from the two instruments, although correlated well, they differed significantly from each other [Lunde 2013]. Using the same health state utility measures 3 months after intracerebral hemorrhage, predictors for lower HRQoL by both scales were higher NIHSS and older age, with similar ORs for EQ-5D-5L, and 15D [Sallinen 2019].”

Q26:Line 337: please say pre-stroke variables – not pre-stroke ‘ones’. Same with post-stroke.

A26: we have changed the text, as suggested.

Q27: Limitations; I would suggest that the selection of the EQ-5D could be a limitation. This instrument is known for its insensitivity for complex and chronic disease. I acknowledge that institutions such as NICE recommend the instrument – the prevalence of an instrument should not drive its selection. Please review the literature on this point and include some points in the Discussion. 

A27: we acknowledge the use of EQ-5D as a limitation. However, we think that using both EQ-5D and 15D adds to the research. On one hand 15D has higher sensitivity and reliability. On the other hand, as EQ-5D is frequently used, our data will be comparable with other research. We have addressed this issue both in the Methods section and the Discussion. 

In the methods section we write:

“While the EQ-5D is widely used [Vainiola 2010], the results might vary depending on the regression method used [Saarni 2006]. EQ-5D seems somewhat less sensitive when it comes to evaluating several chronic diseases [Vartianien 2017; Moock 2008, Kontodimoupoulos 2012, Heiskanen 2016]. The 15D performs better in terms of reliability, discrimination and responsiveness [Sintonen 2001b Arch Hell Med], and performs well after critical illness [Vainiola et al, 2010]. The agreement between the two utility measures was only moderate [Vainiola et al, 2010], and both the mean utilities and the standard deviation differs between EQ-5D and 15D [Richardson et al, 2015]. For these reasons – as recommended by Feeny et al [Feeny et al, 2019] – we decided to use multiple questionnaires in this study.”

In the Discussion we also mention this limitation:

“Fifth, although EQ-5D is widely used, this might be a limitation considering its lower sensitivity in evaluating chronic diseases [Vartianien 2017; Lunde 2013; Kontodimoupoulos 2012; Heiskanen 2016]. For this reason we used15D as well, adding higher sensitivity and reliability for evaluating HRQoL [Sintonen 2001b; Veirtianien 2017; Lunde 2013;, Kontodimoupoulos 2012; Heiskanen 2016].”

Q28: Overall rather than saying worse quality of life you could also say reduced (from population norms too if you investigate and compare the utility values) and diminished from population norms. 

A28: we changed the wording throughout the text, as suggested.

Q29:Also investigate minimal clinically important differences and consider how you can weave both population norms and MCID into the paper to provide further contextualisation to your Results and Discussion sections. 

A29: According to the recommendations of our Reviewer we included the issue of population norms and also of minimal clinically important differences in the manuscript. We devoted two separate paragraphs with several references to these questions:

“Nonetheless we found that the average EQ-5D index in both subgroups (0.65±0.34 for District 8 and 0.75±0.28 for other regions) are reduced compared to the population norm of 0.82 [Janssen and Szende, 2014]. Using EQ-5D the health state utility values after stroke were relatively stable in time: in a systematic review of a total of 199 publications on stroke the median value was 0.63 in studies before 2013 versus 0.65 in studies after 2013 [Betts et al 2020]. For 15D, the difference in health utility values between the socioeconomically disadvantaged district and the other areas also did not reach the level of statistical significance in our study (0.73±0.17 for District 8 and 0.78±0.16 for other regions). These values for 15D are within the range reported for stroke patients with or without vision problems (0.73 – 0.89; [Sand et al, 2015]). It has been reported that EQ-5D and 15D should not be used interchangeably in economic evaluations after stroke, as the utility scores generated from the two instruments, although correlated well, they differed significantly from each other [Lunde 2013]. Using the same health state utility measures 3 months after intracerebral hemorrhage, predictors for lower HRQoL by both scales were higher NIHSS and older age, with similar ORs for EQ-5D-5L, and 15D [Sallinen 2019]. 

Minimally important differences (MIDs) for health state utilities vary by measure and methods and are not well established [Mouelhi et al, 2020]. It has been suggested that for EQ-5D differences among health state utilities of at least 0.036 can be considered clinically important [Feeny 2005]. For EQ-5D, the mean MID among non-stroke patients was 0.074 (range -0.011–0.140) [Walters and Braizer, 2005]. In a Korean version of the EQ-5D-3L questionnaire, MID in stroke patients ranged from 0.08 to 0.12 [Kim 2015], and a similar value of 0.10 was reported by Chen et al [Chen et al 2016]. Considering the population norms of EQ-5D reported for Hungary (0.82, [Janssen and Szende 2014]) the mean state utility value for EQ-5D in our patients (0.65 for District 8 and 0.75 for the wealthier regions) is considerably reduced, i.e. HRQoL is obviously adversely affected in stroke survivors 3 months after the acute event with a signal for reduced HRQoL in District 8. As far as 15D is concerned, there is a lack of information on MID values in stroke patients, and also no population norms are known for Hungary. In other patient groups MID for 15D was estimated between 0.01 – 0.03 [Alanne et al, 2015]. Although by standard statistical methods – similarly to our findings for EQ-5D – the difference between health state utilities by the 15D was not statistically significant, the difference between the mean values of District 8 and other regions (i.e. 0.78 – 0.73 = 0.05) suggests that HRQoL after stroke is reduced more in socioeconomically deprived regions than in wealthier areas.”

Q30:Once again, thank you to the authors and good luck with the next stages of the revisions.

A30: We thank our Reviewers for their recommendations. We indeed feel that the thorough revision made on the basis of requested changes improved our manuscript. 

We hope that the changes we made during revision are satisfactory, and will help the manuscript meet the publication standards in PLOS ONE.

Sincerely yours, 

Ildiko Szocs, MD, PhD

corresponding author

---

## [Decision Letter · Decision Letter 1]

8 Oct 2020

Health related quality of life and satisfaction with care of stroke patients in Budapest: a substudy of the EuroHOPE project

PONE-D-20-05987R1

Dear Dr. Szőcs,

We’re pleased to inform you that your manuscript has been judged scientifically suitable for publication and will be formally accepted for publication once it meets all outstanding technical requirements.

Kind regards,

Seana Gall

Academic Editor

PLOS ONE

Additional Editor Comments (optional):

Reviewers' comments:

Reviewer's Responses to Questions

**Comments to the Author**

1. If the authors have adequately addressed your comments raised in a previous round of review and you feel that this manuscript is now acceptable for publication, you may indicate that here to bypass the “Comments to the Author” section, enter your conflict of interest statement in the “Confidential to Editor” section, and submit your "Accept" recommendation.

Reviewer #1: All comments have been addressed

Reviewer #2: All comments have been addressed

2. Is the manuscript technically sound, and do the data support the conclusions?

Reviewer #1: Yes

Reviewer #2: Yes

3. Has the statistical analysis been performed appropriately and rigorously? 

Reviewer #1: Yes

Reviewer #2: Yes

4. Have the authors made all data underlying the findings in their manuscript fully available?

Reviewer #1: Yes

Reviewer #2: Yes

5. Is the manuscript presented in an intelligible fashion and written in standard English?

Reviewer #1: Yes

Reviewer #2: Yes

6. Review Comments to the Author

Reviewer #1: Thank you. All of my comments have been addressed. Please take another proofreading of the final version particularly the past and present tenses of some sentences.

Reviewer #2: Dear Authors, thank you again for the opportunity to review the article and for your thorough response to the request for revisions - I am satisfied with the revisions. Good luck with the next stages of your article for publication. With kindest regards

7. PLOS authors have the option to publish the peer review history of their article (what does this mean?). If published, this will include your full peer review and any attached files.

Reviewer #1: **Yes: **Hoang Phan

Reviewer #2: No

---

## [Editor Report · Acceptance letter]

13 Oct 2020

PONE-D-20-05987R1 

Health related quality of life and satisfaction with care of stroke patients in Budapest: a substudy of the EuroHOPE project 

Dear Dr. Szőcs:

I'm pleased to inform you that your manuscript has been deemed suitable for publication in PLOS ONE. Congratulations! Your manuscript is now with our production department. 

Kind regards, 

on behalf of

Dr. Seana Gall 

Academic Editor

PLOS ONE